# KDGCN: A Kernel-based Double-level Graph Convolution Network for Semi-supervised Graph Classification with Scarce Labels

## Abstract

Graph classification, which is significant in various fields, often faces the challenge of label scarcity. Under such a scenario, supervised methods based on graph neural networks do not perform well because they only utilize information from labeled data. Meanwhile, semi-supervised methods based on graph contrastive learning often yield complex models as well as elaborate hyperparameter-tuning. In this work, we present a novel semi-supervised graph classification method, which combines GCN modules with graph kernels such as Weisfeiler-Lehman subtree kernel. First, we use a GCN module as well as a readout operation to attain a graph feature vector for each graph in the dataset. Then, we view the graphs as meta-nodes of a supergraph constructed by a graph kernel among graphs. Finally, we use another GCN module, whose inputs are the graph feature vectors, to learn meta-node representations over the supergraph in a semi-supervised manner. Note that the two GCN modules are optimized jointly. Compared to contrastive learning based semi-supervised graph classification methods, our method has fewer hyperparameters and is easier to implement. Experiments on seven benchmark datasets demonstrate the effectiveness of our method in comparison to many baselines including supervised GCNs, label propagation, graph contrastive learning, etc.

## 1 Introduction

Graph classification aims to classify a number of graphs into different classes. It has numerous applications such as judging whether two chemicals belong to the same class or two social relationship patterns are similar. Classical methods for graph classification are usually based on graph kernels (Kriege et al., 2019; Borgwardt & Kriegel, 2005; Vishwanathan et al., 2010; Shervashidze et al., 2011) that are useful tools to quantify the similarity or dissimilarity between graphs. For instance, based on similarity computed by a graph kernel, one can use the k-nearest neighbor method or support vector machines (Cortes & Vapnik, 1995) to classify graphs.

In recent years, Graph Neural Networks (GNNs) (Kipf & Welling, 2017; Velickovic et al., 2018; Hamilton et al., 2017) have shown promising performance in not only node classification but also graph classification (Gilmer et al., 2017; Errica et al., 2020). They iteratively aggregate neighborhood node information and perform graph pooling to attain the classification results. However, GNN strongly relies on neighborhood information and lacks global-level information, since graph pooling strategies are relatively simple and may lead to the over-smooth of the nodes in a certain neighborhood. Moreover, in real graph classification tasks, node labels and graph labels are scarce, resulting in heavy accuracy loss when conducting supervised GNN-based methods since such methods can only utilize information from the labeled data. In contrast, semi-supervised graph classification methods can utilize structural information from the unlabeled data, being consistent with the fact that a graph has abundant structural information such as neighborhood connections and node tags. Differently, some methods apply graph contrastive learning together with label augmentation strategies (You et al., 2020; 2021; Yue et al., 2022). However, graph contrastive learning needs to build up a heuristic feature space where positive samples and negative samples are compared to learn representations, yielding complex models and elaborative hyperparameter-tuning in most of the time; label augmentation may not generate proper labels because it may not learn useful information from a small ratio of labeled data.

In this work, we propose a semi-supervised graph classification method that effectively integrates node-level information and graph-level correlation information via GCN modules (Kipf & Welling, 2017) and graph kernels. The model in our method is composed of two GCN modules. The first one is conducted on each graph of the dataset and provides node-level representations for the graphs and further generates graph-level representations via a readout operation. The second GCN module is conducted on a supergraph constructed by a graph kernel over the graphs and improves the graph-level representations. The two GCNs are optimized jointly, in an semi-supervised manner.

We compare our method with classical and recent graph classification methods on seven datasets when the label rates are 1%, 2%, 10%, and 20%, respectively. The results show that our method has a strong performance on at least six datasets and outperforms other methods on three datasets by a large margin. Moreover, our method reaches high accuracies when the label rates are 1% and 2%, indicating that our method can be applied to real tasks when the labels are scarce.

## 2  RELATED WORK

In this section, we introduce basic concepts of semi-supervised learning as well as three kinds of graph classification methods that are related to our work, including graph kernel, graph neural network, and graph contrastive learning.

### 2.1  SEMI-SUPERVISED LEARNING METHODS

Different from traditional pairwise feature-label training, semi-supervised learning aims to extract information from both labeled and unlabeled data because labels are usually difficult or time-consuming to obtain, as Zhu (2008) illustrated. Semi-supervised learning can be derived in a transductive or inductive way. Transductive methods train models from both the training set and testing set, and infer the labels of the testing set directly. Typical methods include TSVM (Joachims, 2006), GCN (Kipf & Welling, 2017), label propagation algorithms (Raghavan et al., 2007; Yamaguchi et al., 2016; Liu et al., 2019), matrix completion based algorithms (Goldberg et al., 2010; Fan & Chow, 2018), and sparse graph learning (Wang et al., 2022). Meanwhile, inductive methods train models from only the training set and generalize to unseen testing data. Typical methods include harmonic mixture method (Zhu & Lafferty, 2005), graph regularized algorithms (Ando & Zhang, 2006; Belkin et al., 2006) (can also be transductive), GraphSAGE (Hamilton et al., 2017), and graph attention networks (Velickovic et al., 2018; Wang et al., 2019; Zhang et al., 2022).

### 2.2  GRAPH KERNELS

Graph kernel often decomposes a graph into several sub-structures and calculates the kernel value between two graphs based on the inner product of their corresponding sub-structures. After necessary normalization, it can generate a kernel matrix upon $N$ graphs. Then, it can be fed into classifiers such as support vector machines (Cortes & Vapnik, 1995) or semi-supervised label propagation methods (Raghavan et al., 2007; Yamaguchi et al., 2016; Liu et al., 2019) to attain the final classification result. In the following paragraph, we briefly introduce a few typical graph kernels.

Shortest path kernel (Borgwardt & Kriegel, 2005) counts the number of shortest paths that share the same length, starting vertex, and aggregated label for a pair of graphs. Random walk kernel (Vishwanathan et al., 2010) utilizes the Kronecker product to compute a public graph for a pair of graphs and counts the number of random paths. To simplify the computation, Weisfeiler-Lehman (WL) subtree kernel (Shervashidze et al., 2011) iteratively generates multisets based on subtree structure and compresses them into new labels. Then, it outputs the dot product of a pair of graphs based on the count of previous labels and new labels. Moreover, persistent Weisfeiler-Lehman (PWL) kernel (Rieck et al., 2019) fully extracts topology information and generalizes WL subtree kernel. Wasserstein Weisfeiler-Lehman (WWL) kernel (Togninalli et al., 2019) further embeds graphs into continuous distributions and applies Wasserstein distance as the metrics. Differently, isolation graph kernel (Xu et al., 2021) models the distributions of attributed graphs in mean embeddings.

### 2.3 GRAPH NEURAL NETWORKS

Graph Neural Network (GNN) can learn the pattern of a whole graph by iterative aggregation, which integrates neighborhood features for a node and obtains its representation. GCN (Kipf & Welling, 2017) and its variants (Gao et al., 2018; Wu et al., 2019; Abu-El-Haija et al., 2019) pioneerly propagate node embeddings by adjacency matrix and feature matrix in the spectral domain on the whole graph, achieving promising performance in node classification. GraphSAGE (Hamilton et al., 2017) applies inductive learning, which only requires information from $k$-stage neighbors upon the current node. Later, Zhang et al. (2018) proposed an end-to-end framework that directly classifies graphs. Graph Isomorphism Network (GIN) (Xu et al., 2019) theoretically decomposes GNN (e.g. GCN and GraphSAGE) into three steps, including aggregation, combination, and readout. Also, it adapts GCN and GraphSAGE for graph classification and proposes another classification method based on *sum* strategy. Based on GCN, graph attention networks (Velickovic et al., 2018; Wang et al., 2019; Zhang et al., 2022) sort important nodes via the attention mechanism and avoid the over-smooth phenomenon in some cases of GCN. Differently, methods based on graph autoencoder (Pan et al., 2018; Tang et al., 2022; Hou et al., 2022) embed vertices into low-dimensional vectors and reconstruct statistical neighborhood information. Recently, DropGNN (Papp et al., 2021) aggregates results from different GNNs where different nodes will be deleted by a marginal probability. It successfully enhances the robustness of GNN in graph classification. Chauhan et al. (2020) and Huang et al. (2023) constructed context-based supergraphs under few-shot learning and GNN framework.

More recently, a few researchers proposed GNN-based methods for semi-supervised graph classification (Ju et al., 2022c;b;a; 2023). Xie et al. (2022) and Ju et al. (2023) combined active learning with semi-supervision on graph classification. Particularly, Ju et al. (2022c;b;a) integrated GNNs with learnable graph kernel modules in parallel, which showed promising performance in semi-supervised graph classification. Our method is a double-level GCN model and is able to take advantage the hierarchical information of graphs of nodes and graphs of graphs, which is different from (Ju et al., 2022c;b;a).

### 2.4 GRAPH CONTRASTIVE LEARNING

Graph contrastive learning is usually a kind of unsupervised representation learning method (Sun et al., 2019) and can also be applied to semi-supervised learning. It generates positive and negative samples and learns a projection space where similar samples are close to each other and dissimilar samples are far from each other. The GraphCL proposed by (You et al., 2020) learns unsupervised projection space with a combination of four kinds of augmentation strategies. You et al. (2021) further proposed a framework that automatically selects augmentation strategies. Differently, Hassani & Ahmadi (2020) proposed to utilize graph diffusion and maximize the mutual information between different views of a graph, yielding satisfactory results in node-level and graph-level tasks. Later, the SimGRACE proposed by Xia et al. (2022) utilizes perturbed GNN and adversarial training to learn an invariant representation without manual augmentation. More recently, Yue et al. (2022) proposed a method called GLA that utilized a label-invariant augmentation strategy in the projection space and achieved high accuracy in semi-supervised learning when the labels are scarce. Meanwhile, Luo et al. (2022) proposed DualGraph which enhances the consistency of graph contrastive model on unlabeled data via a dual structure.

## 3 PROPOSED METHOD

In this section, we first illustrate the general formulation for our method. Then, we introduce the procedures of our method, including how we determine our two-level models and how we construct the supergraph. Finally, a complexity analysis of our method is carried out.

### 3.1 GENERAL FORMULATION

A graph is usually denoted as $G = (V, E)$, where $V$ denotes the set of vertices and $E$ denotes the set of edges. The corresponding adjacency matrix is denoted as $\mathbf{A} \in \{0, 1\}^{n \times n}$, where $n = |V|$. Sometimes each node of $G$ has a feature vector. The matrix formed by the feature vectors of nodes is denoted as $\mathbf{X} \in \mathbb{R}^{n \times m}$, where $m$ denotes the dimension of the feature. For convenience, we

also denote $G = (\mathbf{A}, \mathbf{X}) \in \mathbb{G}$. Given a set of partially labeled graphs $\mathcal{G} = \mathcal{G}_l \cup \mathcal{G}_u$, where $\mathcal{G}_l = \{G_1, G_2, \ldots, G_l\}$ are labeled as $\{y_1, y_2, \ldots, y_l\}$, $y_i \in [K]$, $\mathcal{G}_u = \{G_{l+1}, G_{l+2}, \ldots, G_{l+u}\}$ are unlabeled, and $u + l = N$, our goal is to predict the labels for $\mathcal{G}_u$. The problem becomes more challenging when $l/u$ is smaller.

Let $h : \mathbb{G} \to \mathbb{R}^d$ be a function that represents a graph $G$ as a $d$-dimensional vector. Denote $\bar{\mathbf{h}}_i = h(G_i)$, $i \in [N]$ and let $\bar{\mathbf{H}} = [\bar{\mathbf{h}}_1^\top; \bar{\mathbf{h}}_2^\top; \ldots; \bar{\mathbf{h}}_N^\top] \in \mathbb{R}^{N \times d}$. Let $s : \mathbb{G} \times \mathbb{G} \to \{0, 1\}$ be a function to determine whether two graphs are similar or dissimilar and denote $\mathbf{S} = [s(G_i, G_j)] \in \{0, 1\}^{N \times N}$. Thus we constructed a super graph $\bar{G} = (\mathbf{S}, \bar{\mathbf{H}})$ for $\mathcal{G}$. In $\bar{G}$, the nodes are the graphs in $\mathcal{G}$ and the edges are given by $\mathbf{S}$. The details about $h$ and $s$ will be introduced later.

Based on $\bar{G}$, we reformulate the original graph classification problem as a semi-supervised node classification problem. First, we predict the labels of $\mathcal{G}$ using the following model:

$$\hat{\mathbf{Y}} = f(\mathbf{S}, \bar{\mathbf{H}}) = f(\mathbf{S}, h(\mathcal{G})), \tag{1}$$

where $\hat{\mathbf{Y}} \in [0, 1]^{N \times K}$, $h(\mathcal{G}) = [h(G_1)^\top; h(G_2)^\top; \ldots; h(G_N)^\top]$, and $f, h$ are learnable. The objective function of the semi-supervised learning is as follows:

$$\mathcal{L}_{f,h} = -\sum_{i=1}^{l} \sum_{k=1}^{K} Y_{ik} \log \hat{Y}_{ik}, \tag{2}$$

where $\mathbf{Y} = [Y_{ik}] \in \{0, 1\}^{l \times K}$ denotes the one-hot encoding matrix of $\{y_1, y_2, \ldots, y_l\}$.

## 3.2 Determining the two-level models $f$ and $h$

In this work, we let $f$ and $h$ be GCNs (Kipf & Welling, 2017) parameterized by $\phi$ and $\psi$ respectively. Specifically, we let

$$f_\phi(\mathbf{S}, \bar{\mathbf{H}}) = \text{Softmax}\left(g_\varphi\left(\hat{\mathbf{S}}\sigma(\hat{\mathbf{S}}\bar{\mathbf{H}}\boldsymbol{\Theta}_1)\boldsymbol{\Theta}_2\right)\right), \tag{3}$$

where $\phi = \{\boldsymbol{\Theta}_1, \boldsymbol{\Theta}_2, \varphi\}$, $\sigma$ is an activation function such as ReLU, the sizes of $\boldsymbol{\Theta}_1$ and $\boldsymbol{\Theta}_2$ are $d \times d'$ and $d' \times d''$, respectively, and $g_\varphi : \mathbb{R}^{d''} \to \mathbb{R}^K$ is a multilayer perception (MLP) parameterized by $\varphi$. Note that $\hat{\mathbf{S}}$ is the normalized self-looped adjacency matrix, i.e.,

$$\hat{\mathbf{S}} = \text{diag}(\mathbf{1}^\top(\mathbf{S} + \mathbf{I}_N))^{-1/2}(\mathbf{S} + \mathbf{I}_N)\text{diag}(\mathbf{1}^\top(\mathbf{S} + \mathbf{I}_N))^{-1/2}. \tag{4}$$

The two-layer GCN module in equation 3 combines the feature of each graph $G_i$ with its neighbors and the MLP $g_\phi$ further enhances the representations of each graph for classification.

To obtain $\bar{\mathbf{H}}$, we apply $h_\psi$ to each graph $G$ in $\mathcal{G}$, i.e.,

$$\mathbf{H}_i = h_\psi(\mathbf{X}_i, \hat{\mathbf{A}}_i) = \hat{\mathbf{A}}_i \sigma(\hat{\mathbf{A}}_i \mathbf{X}_i \mathbf{W}_1)\mathbf{W}_2, \tag{5}$$

where $\psi = \{\mathbf{W}_1, \mathbf{W}_2\}$, $\mathbf{W}_1 \in \mathbb{R}^{m \times d}$, $\mathbf{W}_2 \in \mathbb{R}^{d \times d}$, $\mathbf{X}_i$ is the $n_i \times m$ node feature matrix of $G_i$. $\hat{\mathbf{A}}_i$ is the normalized self-looped adjacency matrix of $G_i$, i.e.,

$$\hat{\mathbf{A}}_i = \text{diag}(\mathbf{1}^\top(\mathbf{A}_i + \mathbf{I}_{n_i}))^{-1/2}(\mathbf{A} + \mathbf{I}_{n_i})\text{diag}(\mathbf{1}^\top(\mathbf{A}_i + \mathbf{I}_{n_i}))^{-1/2}. \tag{6}$$

Finally, the $i$-th column of $\bar{\mathbf{H}}$ is obtained by using a readout function for $\mathbf{H}_i$, e.g.,

$$\bar{\mathbf{h}}_i = \mathbf{H}_i^\top \mathbf{1}, \quad i \in [N]. \tag{7}$$

GCN is simple in form and easy to transplant, and at the same time, it can still perform well when features are scarce. Moreover, GCN can be combined with semi-supervised learning, which extracts the structural information from both labeled and unlabeled data. Thus, GCN modules can be applied to different levels, being consistent with its formula and our motivation. Figure 1 shows the overall structure of our method, where "READOUT" is described in Eq. 7 and MLP stands for the multilayer perceptron described in Eq. 3. We will discuss how we construct the supergraph $\bar{G}$ and the corresponding similarity matrix $\hat{\mathbf{S}}$ in Section 3.3.

The advantages of our method over existing methods are as follows.

- Compared to supervised graph classification methods, our method is able to utilize unlabeled data effectively. This is very important when the labels are scarce.
- Compared to conventional semi-supervised methods such as label-propagation and Laplacian regularization, our method extracts discriminative features via graph convolution and neural networks.
- Compared to semi-supervised graph representation learning methods and graph contrastive learning methods, our method has fewer hyperparameters and is convenient to implement.

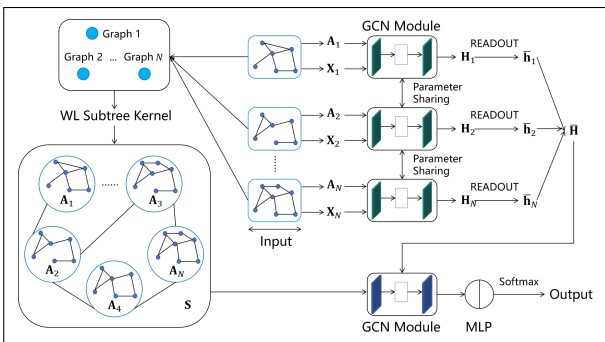

Figure 1: Overall structure of our KDGCN.

## 3.3 CONSTRUCTING THE SUPERGRAPH $\bar{G}$

We use graph kernel to construct the supergraph $\bar{G}$. Firstly, we apply a graph kernel $k(\cdot, \cdot)$ to each pair in $\mathcal{G}$ and form a kernel matrix $\mathcal{K}$, where $\mathcal{K}_{ij} = k(G_i, G_j)$, $(i, j) \in [N] \times [N]$. In addition, we let $\mathcal{K}_{ij} \leftarrow \mathcal{K}_{ij}/(\mathcal{K}_{ii}\mathcal{K}_{jj})^{1/2}$ and then we set $\text{diag}(\mathcal{K}) = \mathbf{0}$. Since $\mathcal{K}$ is usually not sparse and a sparse adjacency matrix often performs better than a dense adjacency matrix, we use the following thresholding operation to obtain a binary adjacency matrix:

$$S_{i,j} = \begin{cases} 0, & \mathcal{K}_{ij} \leq \tau \\ 1, & \mathcal{K}_{ij} > \tau \end{cases} \tag{8}$$

where $0 < \tau < 1$ is a predefined hyperparameter. Rather than using a single threshold for the entire matrix $\mathcal{K}$, one can also consider setting the largest $c$ (an integer) elements in each column of $\mathcal{K}$ to 1 and let the others be zero and obtain a sparse but asymmetric matrix $\bar{\mathcal{K}}$ and let $\mathbf{S} = \max(\bar{\mathcal{K}}, \bar{\mathcal{K}}^\top)$. Intuitively, if $\tau$ is too large or $c$ is too small in the thresholding operation, the supergraph will be too sparse. However, if $\tau$ is too small or $c$ is too large, the supergraph will be a very dense graph and is not discriminative, which leads to low classification accuracy. We can tune this hyperparameter using cross-validation.

As introduced in Section 2.2, there have been many graph kernels. Particularly, the WL kernel family has shown satisfactory performance in a few previous works. WL subtree kernel (Shervashidze et al., 2011), which fully captures the subtree structures of a graph, is based on the Weisfeiler-Lehman test of isomorphism on graphs and hence is a very effective metric of graph similarity. Moreover, the computational cost of the WL subtree kernel is linear with the number of edges and hence is scalable to large datasets (Shervashidze et al., 2011). Therefore, the WL subtree kernel is indeed a strong complement to our two-level GCN model.

In Table 4, we compare the performance of many graph kernels. The results show that the WL subtree kernel is indeed more effective than other graph kernels in many cases. At the same time, we calculate the structural properties of our supergraphs of different datasets to further demonstrate the effectiveness of our supergraph and the thresholding operation, as shown in Appendix D.

## 3.4 COMPLEXITY ANALYSIS

The computational complexity of our method is mainly from the following four parts: (a) the computation of $\hat{\mathbf{S}}$; (b) obtaining $\bar{\mathbf{H}}$ according to equation 5 and equation 7; (c) obtaining $\hat{Y}$ according to

equation 1. For convenience, WLOG, we assume that all graphs in $\mathcal{G}$ have the same number ($n$) of nodes and $d'' = d' = d$. Firstly, it has been shown by (Shervashidze et al., 2011) that the complexity of (a) is $\mathcal{O}(N\tau\bar{s} + N^2\kappa n)$, where $\kappa$ is the iteration of WL subtree kernel, $\bar{s}$ is the number of elements in the multisets of a graph and $n$ is the number of nodes of a graph. The complexity of (b) is $\mathcal{O}(N(n^2d + nmd + nd^2))$. The complexity of (c) is $\mathcal{O}(Nd^2 + N^2d + NLdK)$, where $L \geq 2$ denotes the number of layers in the MLP. Now suppose the total number of iterations in the optimization is $T$, then the total time complexity of our method is $\mathcal{O}(N^2\kappa n + T(Nn^2d + N^2d))$, where we have assumed that $n > \max(m, d)$ and $N > \max(d, LK)$. This complexity can be further reduced if we consider the sparsity of the adjacency matrices and compute the elements of $\mathcal{K}$ in parallel.

## 3.5 EXTENSIONS

Currently, our method KDGCN is based on GCN (Kipf & Welling, 2017) and is only used in transductive learning on graphs. KDGCN can also utilize other GNN modules such as GAT (Velickovic et al., 2018) and GraphSAGE (Hamilton et al., 2017). For example, using the GraphSAGE module, we are able to conduct inductive learning on graphs. The corresponding algorithm is presented in Algorithm 1 of Appendix F.

## 4 EXPERIMENTS

In this section, we first describe seven public graph classification benchmark datasets from TU-Dataset (Morris et al., 2020), evaluation metrics, and experiment settings. Then, we briefly describe baseline algorithms for comparison and show accuracy and standard deviation results under four label rates, including 1%, 2%, 10%, and 20%. Finally, we analyze these results and further demonstrate the effectiveness of our method by exploring the influence of different graph kernels and performing hyperparameter experiments.

### 4.1 DATASETS AND EXPERIMENTAL SETTINGS

Table 1 shows the statistics of these benchmark datasets. The number of graphs ranges from 188 to 5000. Meanwhile, the average number of nodes ranges from 13.00 to 284.32 and the average number of edges ranges from 14.69 to 2457.78. Hence, these datasets can evaluate our method effectively for their diversity. We perform 5-fold cross-validation for model evaluation to all methods and report

Table 1: Statistics of datasets for graph classification tasks.

| Dataset | Category | #Class | #Graph | Average #Node | Average #Edge |
|---------|----------|--------|--------|---------------|---------------|
| MUTAG | Small Molecules | 2 | 188 | 17.93 | 19.79 |
| PTC-MR | Small Molecules | 2 | 344 | 14.29 | 14.69 |
| PROTEINS | Bioinformatics | 2 | 1113 | 39.06 | 72.82 |
| DD | Bioinformatics | 2 | 1178 | 284.32 | 715.66 |
| COLLAB | Social Networks | 3 | 5000 | 74.49 | 2457.78 |
| IMDB-M | Social Networks | 3 | 1500 | 13.00 | 65.94 |
| IMDB-B | Social Networks | 2 | 1000 | 19.77 | 96.53 |

the average and standard deviation of accuracies which are from their best epochs of each fold. For a fair comparison, we apply the same dataset split and the same 5-fold cross-validation strategy to all the comparison methods and attain their results by using their given hyperparameter settings. The settings are described in Appendix A.

### 4.2 ALGORITHMIC PERFORMANCE

We choose WL subtree kernel (Shervashidze et al., 2011), GCN (Kipf & Welling, 2017), GraphSAGE (Hamilton et al., 2017), DGCNN (Zhang et al., 2018), GIN (Xu et al., 2019), MVGRL (Hassani & Ahmadi, 2020), DropGNN (Papp et al., 2021), SimGRACE (Xia et al., 2022) and GLA (Yue et al., 2022) as our comparison algorithms. In our method, We compare two different thresholding operations as described in Section 3.3. Since WL subtree kernel and GCN modules are important components of our method, we can perform an equivalent ablation study by comparing

WL subtree kernel and GCN. Here, WL subtree kernel is combined with the supervised support vector machines (Cortes & Vapnik, 1995) and the semi-supervised label propagation method (Yamaguchi et al., 2016). We denote them as WL+SVM and WL+LP, respectively. Meanwhile, GCN, GraphSAGE, DGCNN, GIN and DropGNN are supervised graph classification methods. Finally, MVGRL is self-supervised while SimGRACE and GLA are semi-supervised. Note that the GCN and the GraphSAGE that we compare are derived from the GIN library because the original codes are for node classification (See Section 2.3).

Table 2 shows the results of each method on the aforementioned datasets in the form of mean accuracy and the corresponding standard deviation when the label rates are 10% and 20%. The highest accuracy is highlighted in red while the second highest accuracy and the third highest accuracy are highlighted in blue. KDGCN-$\tau$ means that the thresholding operation is controlled by $\tau$ while KDGCN-$c$ means that the thresholding operation is controlled by $c$.

Table 2: Graph classification results in the form of Mean Accuracy (%) and Standard Deviation (%) on seven benchmark datasets.

| Label | Methods | MUTAG | PTC-MR | PROTEINS | DD | COLLAB | IMDB-M | IMDB-B |
|---|---|---|---|---|---|---|---|---|
| 10% | WL+SVM | 70.71±1.78 | 52.52±3.11 | 71.72±2.64 | 72.55±1.17 | 72.74±1.72 | 45.47±1.60 | 67.87±3.08 |
| | WL+LP | 66.47±0.00 | 55.81±0.00 | 59.58±0.00 | 58.62±0.00 | 57.36±0.56 | 42.24±1.04 | 64.98±2.57 |
| | GCN | 75.76±2.80 | 56.39±2.45 | 69.74±1.65 | 69.26±0.57 | 73.17±1.48 | 46.64±1.54 | 66.07±3.08 |
| | GraphSAGE | 72.24±0.87 | 58.71±4.46 | 68.76±1.78 | 68.84±1.44 | 70.49±0.79 | 45.56±1.39 | 65.51±2.93 |
| | DGCNN | 82.00±4.64 | 56.84±1.15 | 72.91±0.95 | 76.78±0.31 | 68.23±0.68 | 43.78±1.38 | 59.24±4.77 |
| | GIN | 82.35±6.27 | 56.52±3.37 | 71.94±1.45 | 70.14±1.00 | 73.04±1.07 | 47.50±1.00 | 67.16±3.59 |
| | DropGNN | 82.00±3.89 | 57.61±3.13 | 72.77±0.92 | 75.19±0.72 | 69.71±1.72 | 46.36±1.84 | 69.43±1.77 |
| | MVGRL | 82.24±2.96 | 54.77±2.66 | 69.12±1.46 | 71.63±0.63 | 73.00±1.06 | 43.23±2.93 | 66.76±4.13 |
| | SimGRACE | 83.20±3.79 | 58.32±3.10 | 72.61±1.51 | 74.19±1.29 | 73.87±0.84 | 46.12±1.07 | 68.78±1.04 |
| | GLA | 84.31±3.17 | 56.00±1.49 | 75.04±1.17 | 77.55±1.00 | 74.84±1.16 | 48.15±0.87 | 68.82±3.30 |
| | KDGCN-$\tau$(Ours) | 87.29±2.96 | 61.29±1.56 | 73.85±1.20 | 76.42±0.23 | 74.61±1.18 | 46.27±0.43 | 68.31±2.46 |
| | KDGCN-$c$(Ours) | 89.53±2.47 | 63.16±2.86 | 76.03±0.63 | 77.21±1.93 | 88.88±1.29 | 58.56±2.12 | 86.22±2.15 |
| 20% | WL+SVM | 76.99±2.61 | 58.79±3.08 | 73.11±0.52 | 75.38±0.86 | 75.56±0.72 | 47.18±1.80 | 71.45±1.00 |
| | WL+LP | 66.49±0.22 | 55.81±0.16 | 59.57±0.02 | 58.66±0.04 | 56.00±0.30 | 44.68±1.81 | 69.20±2.14 |
| | GCN | 78.05±2.26 | 61.26±2.93 | 70.93±0.90 | 70.39±1.64 | 75.28±0.53 | 48.05±0.96 | 71.25±1.94 |
| | GraphSAGE | 74.55±5.76 | 61.11±4.09 | 69.13±3.57 | 68.71±3.62 | 73.34±0.95 | 47.53±1.12 | 69.00±1.38 |
| | DGCNN | 84.72±0.98 | 61.55±0.88 | 73.74±0.50 | 77.59±0.65 | 72.27±0.41 | 45.10±1.23 | 65.37±2.43 |
| | GIN | 85.36±1.72 | 60.90±3.52 | 74.07±0.71 | 70.71±1.02 | 75.54±0.60 | 48.65±1.23 | 72.22±1.19 |
| | DropGNN | 82.71±5.24 | 61.63±1.52 | 74.14±0.70 | 75.79±0.85 | 71.66±0.58 | 50.05±0.61 | 72.79±0.88 |
| | MVGRL | 82.72±3.88 | 55.52±1.81 | 70.96±0.94 | 75.91±0.96 | 74.44±0.51 | 47.00±1.62 | 71.78±0.99 |
| | SimGRACE | 86.27±2.91 | 60.58±2.95 | 74.56±0.97 | 76.18±1.60 | 77.57±0.64 | 48.72±1.03 | 71.85±1.53 |
| | GLA | 85.11±3.78 | 58.43±3.04 | 75.34±0.97 | 77.80±0.47 | 77.84±0.27 | 49.08±0.71 | 72.88±0.57 |
| | KDGCN-$\tau$(Ours) | 87.64±0.97 | 62.06±3.62 | 75.18±1.41 | 76.59±0.29 | 77.61±0.67 | 49.60±1.08 | 73.88±1.04 |
| | KDGCN-$c$(Ours) | 91.62±1.79 | 63.81±0.75 | 76.93±1.62 | 77.27±2.11 | 90.63±0.31 | 65.18±1.32 | 91.70±1.07 |

It can be seen from Table 2 that our method reaches the highest mean accuracy on MUTAG, PTC-MR, PROTEINS, COLLAB, IMDB-M and IMDB-B under two label rates, indicating that it can be generalized to different kinds of datasets. Specifically, our method exceeds other methods on social networks datasets (i.e. COLLAB, IMDB-M and IMDB-B) by a large margin and exceeds other methods on small molecules datasets by 2.18% to 5.35%. Moreover, our KDGCN always performs better when utilizing thresholding operation controlled by $c$, especially on social networks datasets. Furthermore, among all experiments, the standard deviations of our method are always under 3.00% and are below 2.00% in most cases, showing that our method is stable and will not derive intense oscillations in mean accuracy under different dataset partitions.

Here, we carry out an analysis on why $c$ is better than $\tau$ and why our KDGCN exceeds a lot in mean accuracy on social networks datasets when utilizing $c$. When constructing the supergraph of our method (See Section 3.3), the thresholding operation regarding $\tau$ tends to build up a centralized and sparse supergraph because $\tau$ is applied globally regardless of the degree or the intrinsic characteristics of every graph (i.e. meta-node). Meanwhile, the operation regarding $c$ sorts the closest neighbors for every meta-node, generating a supergraph whose node degrees are roughly the same. Since social networks datasets are inclined to be dense and decentralized referring to Table 4.1, the thresholding operation regarding $c$ is more consistent with these datasets. At the same time, the operation regarding $c$ can model the intrinsic relationship among meta-nodes better, resulting in higher mean accuracies in all of the above experiments. Statistical analysis on supergraphs constructed via $\tau$ and $c$ is illustrated in Appendix D.

For the comparison with our baseline methods, our KDGCN exceeds the mean accuracy in every experiment against our baseline methods WL+SVM, WL+LP and GCN. The reason for such a phe-

nomenon can be illustrated as follows: (1) The original GCN is designed for node classification, which is not fully compatible with graph classification since it has a weak ability to model the correlation among graphs. (2) Although WL subtree kernel can model the correlation among graphs, it still needs abundant graph labels because the output matrix has to be fed into a downstream classifier. However, datasets such as MUTAG, PTC-MR, and so on do not have enough labels when the label rate is small. At the same time, the above experiments prove that our method successfully utilizes GCN modules to fully extract both node-level information and graph-level information and the WL subtree kernel to generate a similarity matrix between graphs.

### 4.3 PERFORMANCE UNDER EXTREMELY SCARCE LABELS

To further investigate the capability of our method to extract structural information from the datasets where the graph labels are severely scarce, we conduct experiments with five methods that perform well when the label rates are 10% and 20%, including WL+SVM and WL+LP (Shervashidze et al., 2011; Cortes & Vapnik, 1995; Yamaguchi et al., 2016) in Section 4.2, GIN (Xu et al., 2019), DGCNN (Zhang et al., 2018), DropGNN (Papp et al., 2021), SimGRACE (Xia et al., 2022) and GLA (Yue et al., 2022). Here, we set the label rates to be 1% and 2%, and we use the same datasets as well as experiment settings as described in Section 4.1. The results are shown in Table 3, where the highest accuracy is highlighted in red; the second and the third highest accuracy is highlighted in blue. Meanwhile, Table 8 in Appendix C shows the average mean accuracy for the above methods on all datasets when the label rates are 1%, 2%, 10% and 20%, respectively.

Table 3: Graph classification results in the form of Mean Accuracy (%) and Standard Deviation (%) on seven benchmark datasets when the graph labels are extremely scarce.

| Label | Methods | MUTAG | PTC-MR | PROTEINS | DD | COLLAB | IMDB-M | IMDB-B |
|---|---|---|---|---|---|---|---|---|
| 1% | WL+SVM | 60.75±14.14 | 51.14±3.09 | 63.09±4.76 | 65.19±2.05 | 61.48±2.64 | 38.03±3.40 | 56.32±4.49 |
| | WL+LP | 61.08±14.22 | 55.72±0.00 | 59.53±0.00 | 58.69±0.00 | 55.66±2.06 | 38.03±2.54 | 53.21±6.38 |
| | DGCNN | 80.75±4.98 | 56.72±0.68 | 59.53±0.00 | 73.42±3.52 | 62.88±3.71 | 37.74±1.70 | 52.61±2.08 |
| | GIN | 73.39±3.28 | 51.50±4.01 | 63.10±1.90 | 64.30±3.00 | 66.17±2.24 | 40.30±2.59 | 58.85±5.90 |
| | DropGNN | 60.86±14.42 | 49.68±5.10 | 63.61±4.10 | 62.21±2.47 | 60.96±1.33 | 43.13±2.82 | 63.08±3.60 |
| | SimGRACE | 66.02±9.72 | 51.59±1.42 | 68.28±4.95 | 70.91±3.83 | 62.09±4.40 | 37.63±2.87 | 58.61±3.09 |
| | GLA | 70.38±3.32 | 55.95±1.94 | 63.30±6.42 | 74.14±1.09 | 65.09±2.68 | 39.58±1.25 | 61.03±3.67 |
| | KDGCN-$\tau$(Ours) | 83.55±1.89 | 59.71±1.13 | 73.61±1.14 | 76.32±0.22 | 65.91±0.69 | 40.85±2.11 | 60.42±3.52 |
| | KDGCN-$c$(Ours) | 87.42±2.10 | 60.76±1.62 | 70.68±3.31 | 76.30±1.60 | 82.19±1.24 | 44.78±1.08 | 74.44±4.68 |
| 2% | WL+SVM | 67.46±1.05 | 50.65±1.99 | 66.23±3.65 | 69.30±0.71 | 64.33±1.43 | 38.68±3.22 | 62.76±2.90 |
| | WL+LP | 66.49±0.00 | 52.60±4.15 | 59.58±0.00 | 58.70±0.00 | 55.52±1.77 | 37.90±1.90 | 60.29±3.04 |
| | DGCNN | 75.78±4.73 | 58.64±1.50 | 63.89±3.10 | 75.84±0.56 | 65.33±2.32 | 39.39±1.28 | 56.98±2.70 |
| | GIN | 70.05±4.64 | 54.20±2.23 | 64.71±2.67 | 67.15±2.08 | 69.37±0.86 | 40.76±1.61 | 63.47±3.68 |
| | DropGNN | 73.08±3.67 | 55.44±2.25 | 66.70±0.57 | 68.24±2.65 | 62.43±1.86 | 41.84±1.14 | 62.68±2.33 |
| | SimGRACE | 80.54±3.51 | 53.23±0.61 | 69.58±2.75 | 72.63±2.36 | 65.93±3.25 | 39.85±1.43 | 61.71±3.02 |
| | GLA | 73.11±3.43 | 55.98±1.93 | 67.92±3.85 | 74.44±0.67 | 69.33±0.77 | 39.70±1.82 | 65.67±3.49 |
| | KDGCN-$\tau$(Ours) | 83.78±2.51 | 61.78±1.56 | 73.22±0.46 | 76.24±0.18 | 67.24±1.21 | 41.70±2.60 | 65.04±3.16 |
| | KDGCN-$c$(Ours) | 87.14±1.50 | 61.78±1.70 | 73.77±0.61 | 77.73±2.30 | 83.40±2.19 | 47.55±2.26 | 77.73±2.46 |

It can be inferred from Table 3 that our method surpasses other methods on all seven datasets in Section 4.1, proving that our method is robust even when the labels are extremely scarce. Specifically, our method exceeds other methods on COLLAB and IMDB-B by a large margin, being consistent with the fact that the thresholding operation regarding $c$ can construct better supergraphs for social networks datasets. Moreover, our KDGCN exceeds other methods on the rest five datasets by 1.65% to 6.67%. Furthermore, the thresholding operation regarding $c$ still performs better than that regarding $\tau$ generally. On the contrary, other methods do not perform well in such case. The reason lies in two aspects: (1) Supervised methods including DGCNN, GIN and DropGNN rely on labeled graphs and cannot utilize structural information from unlabeled graphs. However, labeled graphs are extremely scarce. (2) Methods based on graph contrastive learning including SimGRACE and GLA still need enough labels to generate a credible representation space. Otherwise, the representation space may be biased. Finally, the standard deviations are mostly less than 3.00% among the above experiments, fully demonstrating the robustness of our method in extreme cases.

### 4.4 INFLUENCE OF GRAPH KERNELS

To further investigate how different graph kernels influence our method, we replace WL subtree kernel with shortest path kernel (Borgwardt & Kriegel, 2005), graphlet sampling kernel (Shervashidze et al., 2009), propagation kernel (Neumann et al., 2016), pyramid match kernel (Nikolentzos et al.,

2017), neighborhood hash kernel (Hido & Kashima, 2009) and WWL kernel Togninalli et al. (2019), respectively on MUTAG, PROTEINS and IMDB-M datasets (Morris et al., 2020). These three datasets belong to small molecules dataset, bioinformatics dataset and social network dataset, respectively. For each dataset, the left column stands for the results where $\tau$ is applied to control the thresholding operation while the right column stands for the results where $c$ is applied, as described in Section 3.3.We utilize the same methodology as Section 3 illustrates and the same evaluation metrics as Section 4.1 illustrates. The results are shown in Table 4, where the label rates are 10% and 20%, respectively. The highest and second-highest accuracies are highlighted in red and blue.

Table 4: Graph classification results in the form of Mean Accuracy (%) and Standard Deviation (%) on MUTAG, PROTEINS and IMDB-M using different graph kernels.

| Label | Kernels | MUTAG | | PROTEINS | | IMDB-M | |
|---|---|---|---|---|---|---|---|
| 10% | shortest path kernel | 85.88±4.30 | 91.53±1.07 | 73.55±0.99 | 76.89±0.56 | 46.74±1.02 | 53.53±1.08 |
| | graphlet sampling kernel | 67.06±0.72 | 82.47±2.80 | 69.26±0.81 | 86.85±6.92 | 39.67±1.15 | 60.31±1.53 |
| | propagation kernel | 82.94±3.28 | 87.53±1.83 | 74.29±1.05 | 73.13±1.41 | 46.27±0.43 | 54.00±1.49 |
| | pyramid match kernel | 85.65±1.07 | 94.00±2.29 | 74.27±0.95 | 76.95±1.02 | 42.80±1.71 | 53.26±0.95 |
| | neighborhood hash kernel | 86.12±1.07 | 92.12±1.98 | 73.39±1.28 | 76.53±0.39 | 46.27±0.43 | 53.30±2.06 |
| | WWL kernel | 86.82±3.18 | 84.59±2.89 | 71.30±1.10 | 75.93±0.64 | 45.10±0.94 | 57.39±1.99 |
| | WL subtree kernel | 87.29±2.96 | 89.53±2.47 | 73.85±1.20 | 76.03±0.63 | 46.27±0.43 | 58.56±2.12 |
| 20% | shortest path kernel | 86.04±1.94 | 93.75±1.19 | 75.00±0.70 | 77.11±1.05 | 48.38±0.83 | 57.52±1.50 |
| | graphlet sampling kernel | 67.42±0.86 | 83.25±1.34 | 69.07±0.87 | 89.01±8.91 | 39.45±1.19 | 68.45±2.54 |
| | propagation kernel | 84.18±0.76 | 88.96±1.56 | 74.82±1.73 | 75.22±1.98 | 49.60±1.08 | 59.57±0.80 |
| | pyramid match kernel | 85.11±1.61 | 95.08±1.60 | 75.45±0.93 | 76.57±1.26 | 43.38±0.89 | 59.92±2.57 |
| | neighborhood hash kernel | 85.24±2.02 | 94.15±0.05 | 75.25±1.48 | 76.80±0.38 | 49.60±1.08 | 57.32±2.96 |
| | WWL kernel | 86.04±1.69 | 86.97±2.36 | 72.04±0.85 | 76.08±1.44 | 46.62±1.58 | 62.97±1.59 |
| | WL subtree kernel | 87.64±0.97 | 91.62±1.79 | 75.18±1.41 | 76.93±1.62 | 49.60±1.08 | 65.18±1.32 |

Table 4 shows that when utilizing WL subtree kernel which is guided by the threshold $\tau$, our KDGCN performs the best on MUTAG under two label rates and performs the best on IMDB-M when the label rate is 20%. Meanwhile, when the threshold switches to $c$, our KDGCN can also perform well on IMDB-M as well as reach fair accuracies on MUTAG and PROTEINS when using WL subtree kernel. Among all kernels, WL subtree kernel leads to the highest accuracy or the second highest accuracy in most of the experiments. Although graphlet sampling kernel performs the best on PROTEINS and IMDB-M when guided by $c$, it does not perform well in other experiments, indicating that it is not as stable as WL subtree kernel. In general, Table 4 proves that WL subtree kernel is solid in building up supergraphs on different kinds of datasets. At the same time, graphlet sampling kernel, propagation kernel, pyramid match kernel and neighborhood hash kernel enjoy fair performances, indicating that our method is compatible with different kernels.

Here, we carry out an analysis on why we choose WL subtree kernel. Firstly, graphlet sampling kernel and WWL kernel are far more time-consuming than WL subtree kernel since enumerating graphlets of different sizes and calculating Wasserstein distance in respective of these kernels take up much time. The time cost for each graph kernel is listed in Appendix E. Secondly, WL subtree kernel iteratively captures topologic structures of different depths from a graph, while GCN module iteratively passes message to nodes of different hops, indicating that WL subtree kernel is consistent with GCN module intrinsically (Xu et al., 2019).

## 4.5 HYPERPARAMETER EXPERIMENTS

We explore how the threshold $\tau$ and $c$ in Section 3.3 influence the classification accuracy. We carry out experiments for all datasets under several thresholds when the label rates are 10% and 20%, respectively. The results and analysis regarding $\tau$ and $c$ are illustrated in Appendix B.

## 5 CONCLUSION

This work has introduced a semi-supervised graph classification method called KDGCN, which integrates GCN modules on different levels and WL subtree kernel to capture sufficient structural information from labeled and unlabeled graphs. Extensive experiments showed that our method is concise in form and competent against other methods when the graph labels are scarce. Moreover, we carry out hyperparameter experiments, which further validate the effectiveness of GCN modules and WL subtree kernel.

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

## A  Experiment Settings

When conducting our KDGCN, the dropout rate is set to 0, the learning rate is set to 0.01 and the weight decay is set to 0.05. We set $d = d' = d'' = 64$ for IMDB-B while we set $d = d' = 64$ and $d'' = 16$ for the rest 6 datasets, as described in Section 3.2. Moreover, for PROTEINS, the node feature matrix $\mathbf{X}$ in Section 3.2 is constructed by one-hot representations of the node labels; for the rest 6 datasets, $\mathbf{X}$ is constructed by one-hot representations of the node degrees. The thresholds $\tau$ and $c$ of each dataset are illustrated in Table 5. The above hyperparameters are chosen by grid search on small datasets derived from the original datasets.

Table 5: Thresholds $\tau$ and $c$ for each dataset.

| Dataset | $\tau$ | $c$ |
|---|---|---|
| MUTAG | 0.95 | 7 |
| PTC-MR | 0.95 | 2 |
| PROTEINS | 0.95 | 28 |
| DD | 0.95 | 19 |
| COLLAB | 0.98 | 42 |
| IMDB-M | 0.95 | 18 |
| IMDB-B | 0.95 | 30 |

## B  Hyperparameter Experiments

Figure 2 shows the influence of $\tau$ and $c$ on the classification accuracy for small molecules datasets, as mentioned in Section 4.1 and Section 4.5. Here, "MA" stands for mean accuracy, "Identity" means that the matrix $\hat{\mathbf{S}}$ in Section 3.2 is exactly an identity matrix, "tau" stands for $\tau$, "c" stands for $c$, "0.1" means the label rate is 10% and "0.2" means the label rate is 20%. Meanwhile, Figure 3 shows the influence of $\tau$ and $c$ for bioinformatics datasets and Figure 4 shows the influence of $\tau$ and $c$ for social networks datasets. When conducting the above experiments, other settings are the same as what Appendix A has illustrated.

As Figure 2, 3 and 4 show, the mean accuracy generally increases with the rise of $\tau$ and with the fall of $c$. After reaching the turning point, the mean accuracy decreases in most cases. Also, the thresholding operation regarding $c$ performs a lot better than that regarding $\tau$ in Section 3.3.

Here, we carry out an analysis. Firstly, as $\tau$ increases and $c$ decreases, the number of graphs that are "adjacent" according to $\mathbf{S}$ in Section 3.2 and 3.3 decreases. That implies WL subtree kernel (Shervashidze et al., 2011) has a high tolerance on graphs that are not homogeneous. In other words, if $\tau$ is too low or $c$ is too high, graphs that are not actually "adjacent" will be noisy information, which impedes our model. However, if $\tau$ is too high or $c$ is too low, there will be insufficient adjacency information for our model to learn. Therefore, to better fulfill the ability of $\mathbf{S}$, an appropriate $\tau$ or $c$ is needed. Moreover, as $c$ dereases in Figure 4, the mean accuracies drastically decrease for social networks datasets, indicating that the supergraphs for social networks datasets should not be sparse. Furthermore, if $\hat{\mathbf{S}}$ equals to an identity matrix, the mean accuracy is still relatively high, indicating that the GCN modules (Kipf & Welling, 2017) that we apply have high potential in both node-level information extraction and graph classification. To sum up, the above experiments further testify that both the GCN module and WL subtree kernel contribute a lot to our method.

We also explore how $d$, $d'$ and $d''$ influence the effectiveness of our method. We choose MUTAG, PROTEINS and IMDB-M (Morris et al., 2020) as the comparative datasets where the label rates are

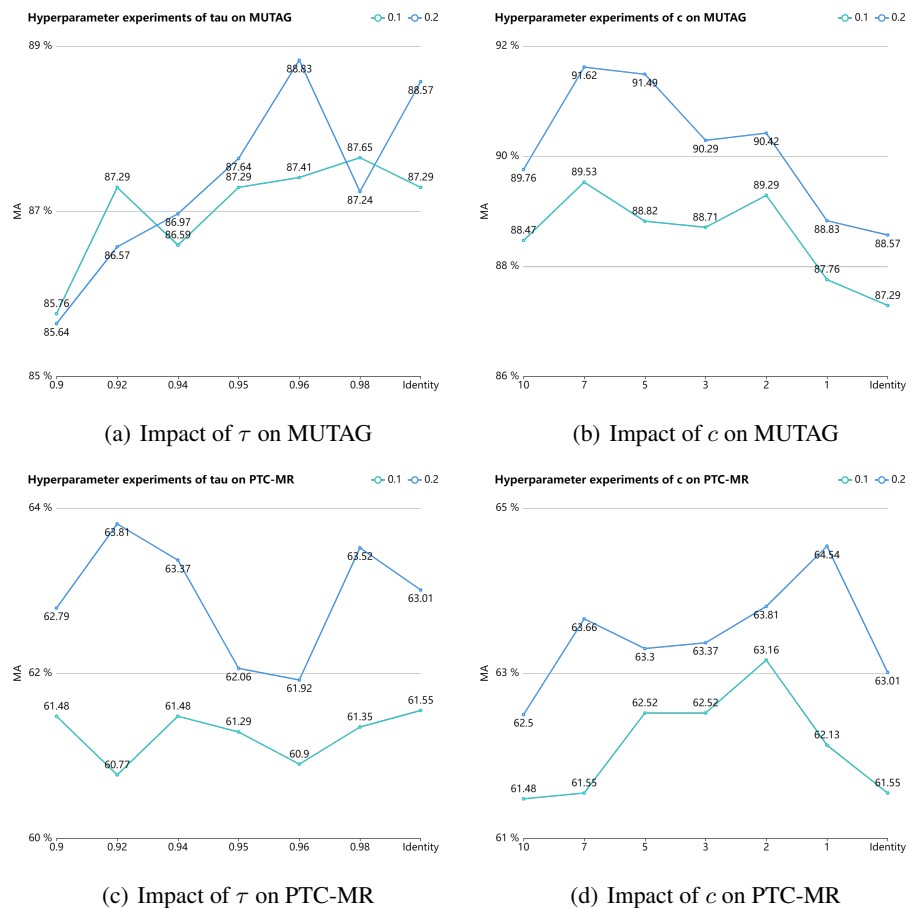

Figure 2: Impact of $\tau$ and $c$ on the mean accuracies of small molecules datasets under different label rates.

set to 10% and 20%, respectively. Firstly, we let $d'' = 16$ and choose different $d$ and $d'$, where $d = d'$ for convenience. Then, we let $d = d' = 64$ and choose different $d''$. Here, the thresholding operation is the one regarding $c$ and other settings are the same as Appendix A has illustrated. The results are shown in Table 6 and 7, respectively, where the highest mean accuracy is highlighted in red and the second highest mean accuracy is highlighted in blue.

The results indicate that when choosing $d = d' = 64$ and $d'' = 16$ on these datasets, our method has a fair performance. Meanwhile, our method does not perform well on IMDB-M when $d, d'$ and $d''$ are too small. However, $d, d'$ and $d''$ that are too large will increase the computation burden as well as lead to sub-optimal results. Thus, moderate $d, d'$ and $d''$ fit for our method.

## C   AVERAGE MEAN ACCURACY FOR DIFFERENT METHODS

In order to quantify the average ability of each method in Table 3 on all datasets described in Section 4.1, we caluculate the average value of the mean accuracies of all datasets for each method under different label rates, as shown in Table 8. The highest average value will be highlighted in red while the second and the third highest average value will be highlighted in blue.

Table 8 shows that when utilizing thresholding operation regarding $c$, our method surpasses other methods on the average value by 7.84% to 9.59%; when utilizing thresholding operation regarding $\tau$, our methods exceeds other methods by 0.48% to 4.42%. Specifically, our method performs the best when the label rate is 1%, proving the robustness of our method under extreme label scarcity. In

Table 6: Graph classification results in the form of Mean Accuracy (%) and Standard Deviation (%) on MUTAG, PROTEINS and IMDB-M where $d''$ is fixed and different $d$ and $d'$ are chosen.

| Label | $d = d'$ | MUTAG | PROTEINS | IMDB-M |
|---|---|---|---|---|
| 10% | 8 | 89.76±2.87 | 76.25±2.25 | 56.16±3.37 |
| | 16 | 89.06±2.45 | 76.35±2.11 | 57.56±2.07 |
| | 32 | 88.71±2.47 | 75.61±1.14 | 59.14±3.30 |
| | 64 | 89.53±2.47 | 76.03±0.63 | 58.56±2.12 |
| | 128 | 90.47±2.54 | 75.07±1.70 | 60.03±1.73 |
| | 256 | 89.29±2.80 | 75.31±1.67 | 59.66±0.96 |
| 20% | 8 | 91.89±2.44 | 76.03±0.88 | 60.97±3.63 |
| | 16 | 90.69±3.11 | 76.41±1.34 | 63.97±2.92 |
| | 32 | 91.49±2.74 | 76.75±2.26 | 66.53±2.01 |
| | 64 | 91.62±1.79 | 76.93±1.62 | 65.18±1.32 |
| | 128 | 91.49±1.59 | 75.25±1.19 | 66.57±1.95 |
| | 256 | 92.16±2.07 | 76.57±1.45 | 64.67±2.91 |

Table 7: Graph classification results in the form of Mean Accuracy (%) and Standard Deviation (%) on MUTAG, PROTEINS and IMDB-M where $d$ and $d'$ are fixed and different $d''$ is chosen.

| Label | $d''$ | MUTAG | PROTEINS | IMDB-M |
|---|---|---|---|---|
| 10% | 4 | 89.65±1.35 | 76.37±1.96 | 58.55±2.74 |
| | 8 | 89.53±2.92 | 75.63±0.92 | 57.99±1.20 |
| | 16 | 89.53±2.47 | 76.03±0.63 | 58.56±2.12 |
| | 32 | 90.24±2.81 | 75.85±0.92 | 57.76±1.78 |
| | 64 | 90.59±1.86 | 75.57±1.85 | 59.10±2.01 |
| | 128 | 89.29±2.29 | 74.73±1.24 | 57.73±3.71 |
| 20% | 4 | 90.69±3.41 | 77.38±2.03 | 64.23±1.72 |
| | 8 | 91.36±2.85 | 76.10±2.51 | 66.30±2.02 |
| | 16 | 91.62±1.79 | 76.93±1.62 | 65.18±1.32 |
| | 32 | 91.09±2.74 | 76.24±1.37 | 65.80±1.84 |
| | 64 | 92.55±1.85 | 76.50±1.90 | 66.30±2.33 |
| | 128 | 91.23±2.85 | 75.74±1.69 | 66.33±2.30 |

Table 8: Average mean accuracy (%) for different methods when the label rates are 1%, 2%, 10% and 20%, respectively.

| Methods | 1% | 2% | 10% | 20% |
|---|---|---|---|---|
| WL+SVM | 56.57 | 59.92 | 64.80 | 68.35 |
| WL+LP | 54.56 | 55.87 | 57.87 | 58.63 |
| DGCNN | 60.52 | 62.26 | 65.68 | 68.62 |
| GIN | 59.66 | 61.39 | 66.95 | 69.64 |
| DropGNN | 57.65 | 61.49 | 67.58 | 69.82 |
| SimGRACE | 59.30 | 63.35 | 68.16 | 70.82 |
| GLA | 61.35 | 63.74 | 69.24 | 70.93 |
| KDGCN-$\tau$ | 65.77 | 67.00 | 69.72 | 71.79 |
| KDGCN-$c$ | 70.94 | 72.73 | 77.08 | 79.59 |

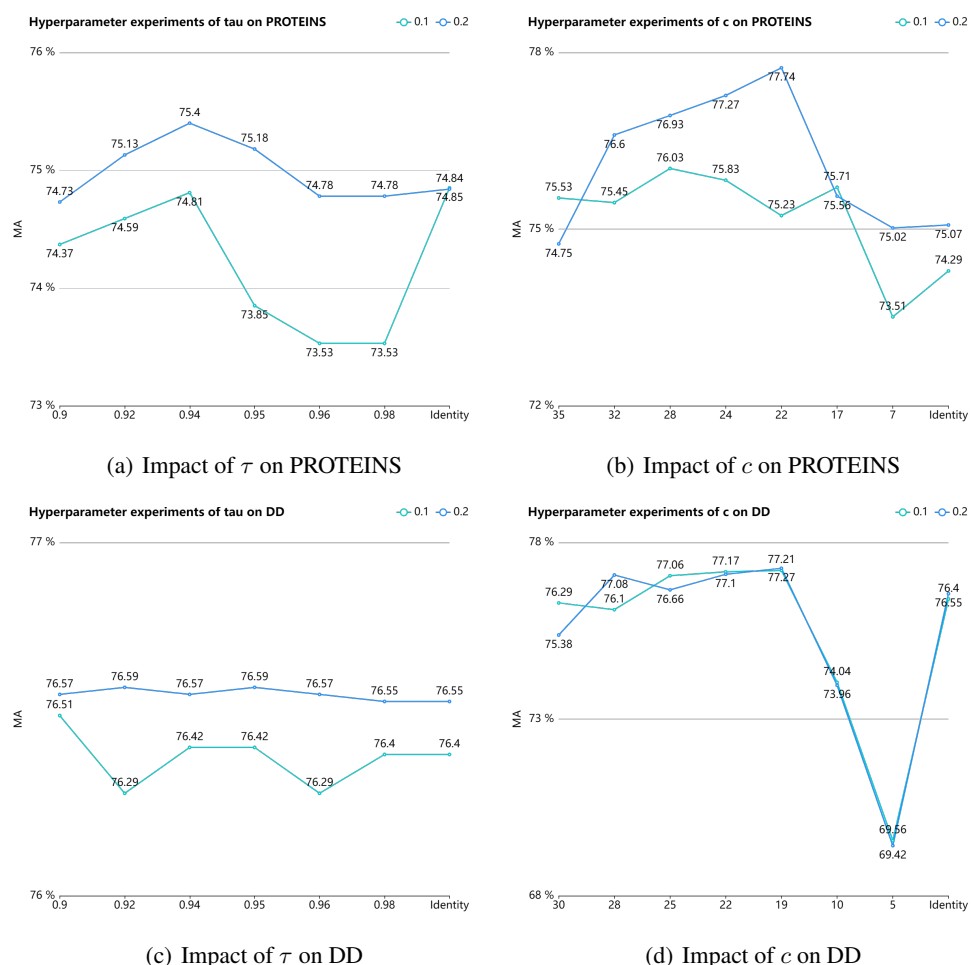

Figure 3: Impact of $\tau$ and $c$ on the mean accuracies of bioinformatics datasets under different label rates.

general, Table 8 further shows that our KDGCN can generalize to different kinds of datasets under different label rates.

## D    STRUCTURAL PROPERTIES OF OUR SUPERGRAPHS

In this section, we first compare the structural properties of **S** in Section 3.3 for all datasets in Section 4.1, where two different thresholding operations regarding $\tau$ and $c$ are applied, respectively. The values of $\tau$ and $c$ for different datasets are the same as Table 5 illustrates. The results are shown in Table 9, where average node degree, number of connected components, average clustering coefficient, average degree centrality and average closeness centrality are the structural properties that we compare. They are denoted as AND, CON, ACCO, ADC and ACCE, respectively.

Table 9 demonstrates that when utilizing $\tau$, the supergraphs tend to be sparse and centralized since the average node degree and the average closeness centrality for each dataset are generally small while the average clustering coefficient and the average degree centrality are large. Here, the larger the average closeness centrality, the easier a meta-node in the supergraph to reach other meta-nodes. Meanwhile, when utilizing $c$, the supergraphs tend to be dense and decentralized. Moreover, when utilizing $\tau$, the supergraphs are inclined to be made up of cliques or isolated meta-nodes while the supergraphs are inclined to be connected when utilizing $c$. This may be the reason why thresholding operation regarding $c$ is better than that regarding $\tau$ in most cases, since a connected supergraph can

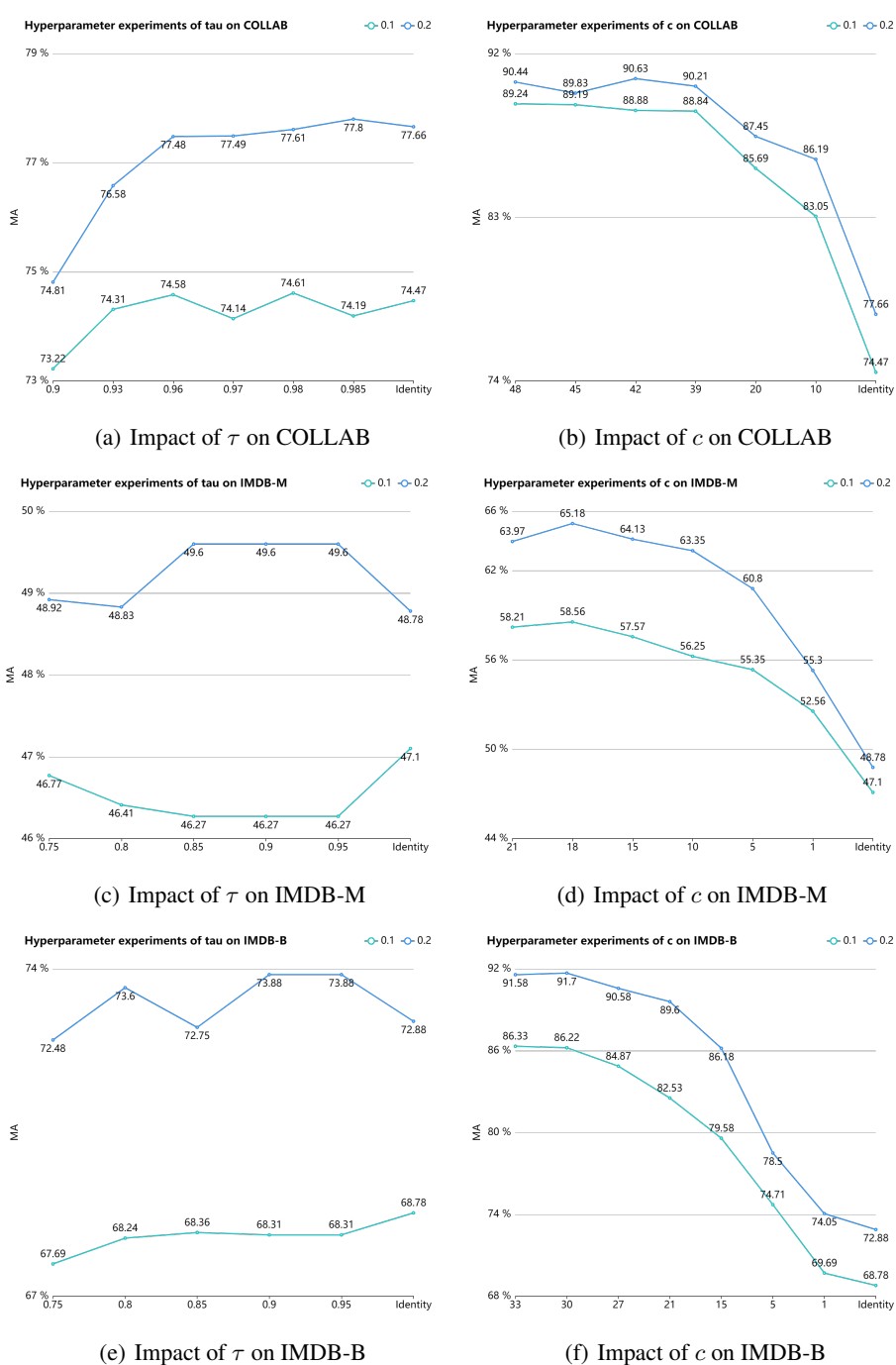

Figure 4: Impact of $\tau$ and $c$ on the mean accuracies of social networks datasets under different label rates.

provide more topological information. Meanwhile, a supergraph made up of several cliques cannot fully propagate useful messages because the relationship among these cliques is still unknown, even though such supergraph can strengthen the message passing of several central meta-nodes.

Then, we perform a case study on PROTEINS (Morris et al., 2020) to investigate how the above structural properties change with the change of $\tau$ and $c$, as shown in Table 10 and Table 11, respectively.

Table 9: Structural properties of the supergraphs of different datasets under different thresholding operations.

| Thresholding | Properties | MUTAG | PTC-MR | PROTEINS | DD | COLLAB | IMDB-M | IMDB-B |
|---|---|---|---|---|---|---|---|---|
| | AND | 1.43 | 0.09 | 0.25 | 0.005 | 5.67 | 95.79 | 6.71 |
| | CON | 17 | 14 | 24 | 1 | 158 | 99 | 116 |
| $\tau$ | ACCO (%) | 41.36 | 10.34 | 61.84 | 100.00 | 85.94 | 93.23 | 80.66 |
| | ADC (%) | 4.35 | 3.94 | 4.81 | 100.00 | 2.39 | 9.79 | 2.01 |
| | ACCE (%) | 9.50 | 3.94 | 4.82 | 100.00 | 2.39 | 9.79 | 2.01 |
| | AND | 13.40 | 3.90 | 55.01 | 37.63 | 83.52 | 35.71 | 58.91 |
| | CON | 1 | 1 | 1 | 1 | 1 | 1 | 1 |
| $c$ | ACCO (%) | 29.15 | 19.57 | 36.49 | 39.94 | 39.65 | 29.15 | 46.10 |
| | ADC (%) | 7.17 | 1.14 | 4.95 | 3.20 | 1.67 | 2.38 | 5.90 |
| | ACCE (%) | 40.53 | 21.25 | 45.28 | 48.69 | 44.38 | 38.98 | 48.98 |

Table 10: Structural properties of the supergraph of PROTEINS when utilizing $\tau$ of different values.

| Properties | 0.90 | 0.92 | 0.94 | 0.95 | 0.96 | 0.98 |
|---|---|---|---|---|---|---|
| AND | 2.42 | 0.81 | 0.31 | 0.25 | 0.23 | 0.23 |
| CON | 23 | 25 | 27 | 24 | 19 | 19 |
| ACCO (%) | 59.94 | 51.57 | 52.67 | 61.84 | 68.75 | 68.75 |
| ADC (%) | 2.74 | 2.40 | 2.92 | 4.81 | 6.40 | 6.40 |
| ACCE (%) | 16.68 | 8.87 | 3.41 | 4.82 | 6.40 | 6.40 |

Table 11: Structural properties of the supergraph of PROTEINS when utilizing $c$ of different values.

| Properties | 35 | 32 | 28 | 22 | 17 | 7 |
|---|---|---|---|---|---|---|
| AND | 68.51 | 62.76 | 55.01 | 43.35 | 33.58 | 13.91 |
| CON | 1 | 1 | 1 | 1 | 1 | 1 |
| ACCO (%) | 38.44 | 37.65 | 36.49 | 35.00 | 33.52 | 30.08 |
| ADC (%) | 6.16 | 5.64 | 4.95 | 3.90 | 3.02 | 1.25 |
| ACCE (%) | 47.31 | 46.50 | 45.28 | 43.08 | 40.97 | 33.65 |

Table 10 shows that the supergraph is sparser and more centralized with the rise of $\tau$, since the average node degree and the average closeness centrality gradually decline and the average clustering coefficient and the average degree centrality gradually increase in such process. At the same time, Table 11 shows that the supergraph is sparser and more decentralized with the fall of $c$, since the average closeness centrality, the average node degree, the average clustering coefficient and the average degree centrality gradually decline in such process. Besides, the supergraph generated according to $c$ has a relatively large centrality because its average clustering coefficient and average degree centrality is relatively high. In consequence, if we choose an adequate $c$, we will attain a supergraph which maintains fair connectivity and density as well as a certain centrality, although not as centralized as that controlled by $\tau$. As a result, such supergraph can not only build up solid message passing but also emphasize important meta-nodes, further proving that the thresholding operation regarding $c$ is better than that regarding $\tau$, especially on social networks datasets which tend to be dense and contain several central nodes.

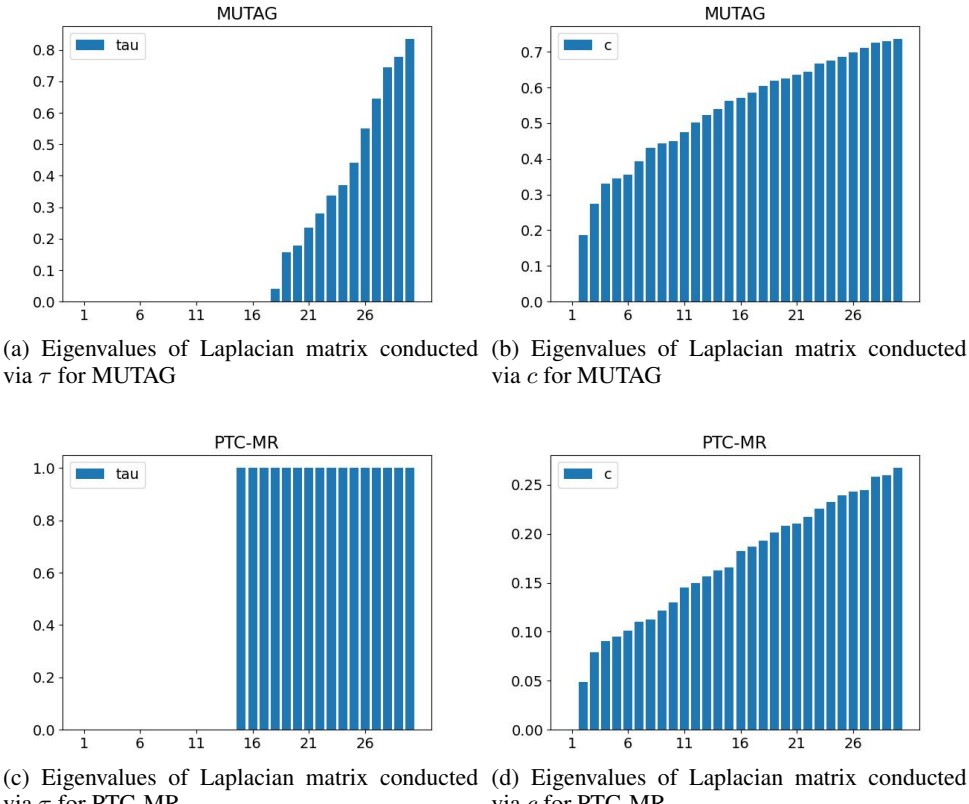

(a) Eigenvalues of Laplacian matrix conducted via $\tau$ for MUTAG

(b) Eigenvalues of Laplacian matrix conducted via $c$ for MUTAG

(c) Eigenvalues of Laplacian matrix conducted via $\tau$ for PTC-MR

(d) Eigenvalues of Laplacian matrix conducted via $c$ for PTC-MR

Figure 5: Top 30 smallest eigenvalues of Laplacian matrices for small molecules datasets under different thresholding oeprations.

Finally, we calculate the Laplacian matrix $\mathbf{L}$ of the supergraph $\mathbf{S}$ for each dataset where $\tau$ and $c$ of each dataset are chosen according to Table 5, as shown in Eq. 9:

$$\mathbf{L} = \mathbf{I}_N - \mathbf{D}^{-1/2}\mathbf{S}\mathbf{D}^{-1/2}, \tag{9}$$

where $\mathbf{D}$ is the degree matrix for $\mathbf{S}$. We calculate the eigenvalues of $\mathbf{L}$ and sort them in ascending order. Then, we select top $z$ smallest eigenvalues for each dataset and plot the bar charts, where $\mathbf{S}$ is conducted in $\tau$ and $c$, respectively. We let $z = 140$ for social networks datasets and $z = 30$ for the rest datasets. The results are shown in Figure 5, 6 and 7, where in each sub-figure, the horizontal axis is the sequence and the vertical axis denotes the top $z$ smallest eigenvalues in ascending order. Meanwhile, the sub-figures in the left column are derived from $\mathbf{S}$ conducted via $\tau$ while those in the right column are derived from $\mathbf{S}$ conducted via $c$.

Figure 5, 6 and 7 show that in all datasets, there are several eigenvalues that equal to 0 when $\mathbf{S}$ is conducted by $\tau$, meaning that such supergraph contains several disconnected cliques. Meanwhile, the first non-zero eigenvalue for the Laplacian matrix conducted via $\tau$ is generally higher than that conducted via $c$, indicating that $\mathbf{S}$ generated by $\tau$ is more centralized than that generated by $c$.

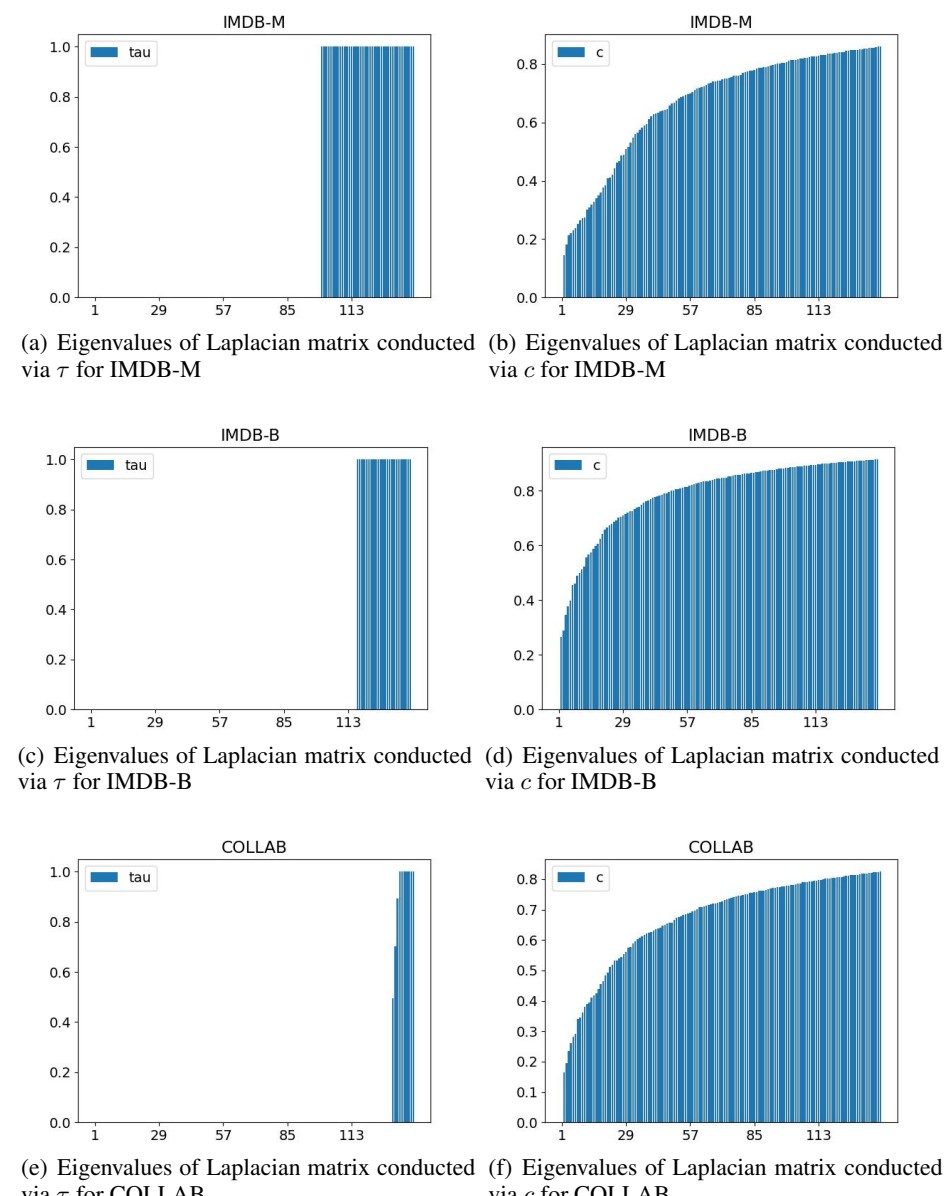

(a) Eigenvalues of Laplacian matrix conducted via $\tau$ for IMDB-M

(b) Eigenvalues of Laplacian matrix conducted via $c$ for IMDB-M

(c) Eigenvalues of Laplacian matrix conducted via $\tau$ for IMDB-B

(d) Eigenvalues of Laplacian matrix conducted via $c$ for IMDB-B

(e) Eigenvalues of Laplacian matrix conducted via $\tau$ for COLLAB

(f) Eigenvalues of Laplacian matrix conducted via $c$ for COLLAB

Figure 6: Top 140 smallest eigenvalues of Laplacian matrices for social networks datasets under different thresholding oeprations.

## E  TIME COST COMPARISON

Firstly, we compare the time cost of different graph kernels in Section 4.4 which construct the supergraphs of PROTEINS, IMDB-M and COLLAB (Morris et al., 2020). These datasets are relatively large in scale and can fully test the capability of the above graph kernels. The results are shown in Table 12, where the time cost is denoted in minute, "-" means that such kernel leads to memory

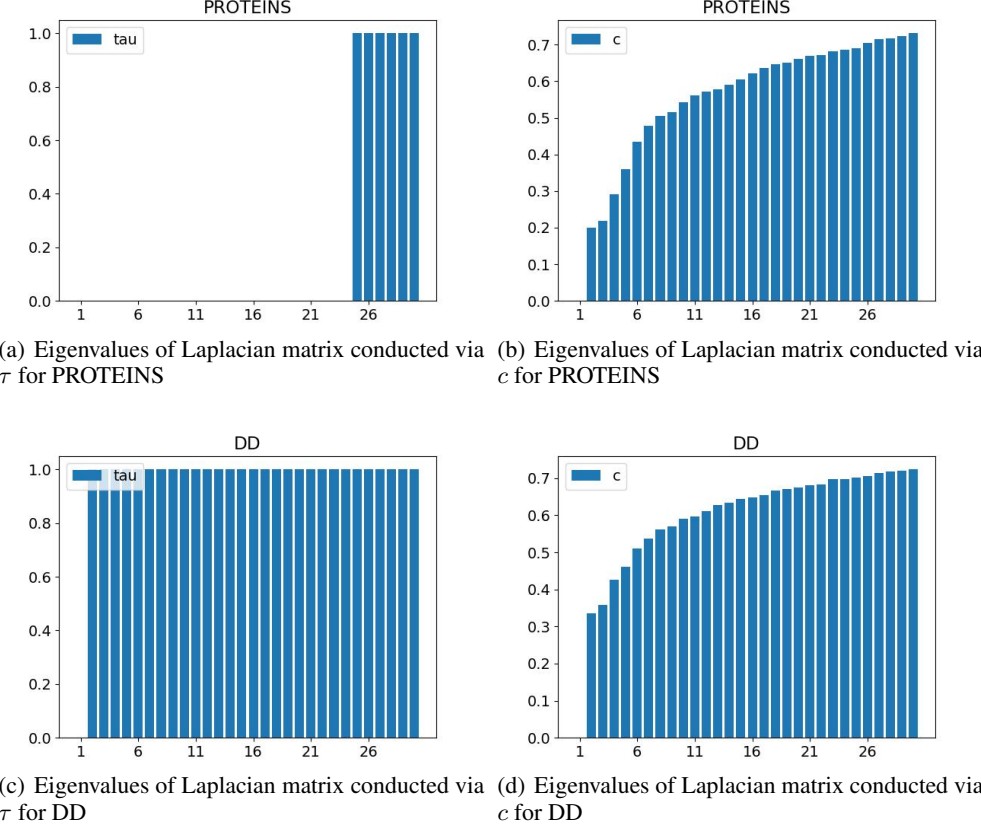

(a) Eigenvalues of Laplacian matrix conducted via $\tau$ for PROTEINS

(b) Eigenvalues of Laplacian matrix conducted via $c$ for PROTEINS

(c) Eigenvalues of Laplacian matrix conducted via $\tau$ for DD

(d) Eigenvalues of Laplacian matrix conducted via $c$ for DD

Figure 7: Top 30 smallest eigenvalues of Laplacian matrices for bioinformatics datasets under different thresholding oeprations.

overflow, the lowest time cost is highlighted in red and the second lowest time cost is highlighted in blue for each dataset.

Table 12 shows that WL subtree kernel has the least time cost on PROTEINS and COLLAB and the second least time cost on IMDB-M, proving that WL subtree kernel is able to handle large-scale datasets efficiently. Also, shortest path kernel performs rather fast on these datasets. Referring to Table 4, it can be concluded that WL subtree kernel can reach the highest accuracy or the second highest accuracy in most cases while consuming litte time. Although graphlet sampling kernel gets the highest accuracy on PROTEINS and IMDB-M when utilizing the thresholding operation regarding $c$, its time cost is huge and will trigger out-of-memory problem when handling large-scale datasets. To sum up, Table 4 and 12 further proves that WL subtree kernel is better than other graph kernels considering accuracy as well as time cost.

Moreover, we compare the average inference time of all epochs and the memory usage for DGCNN (Zhang et al., 2018), GIN (Xu et al., 2019), DropGNN (Papp et al., 2021), GLA (Yue et al., 2022) and our KDGCN-$c$ on PROTEINS, IMDB-M and COLLAB. The label rate is set to 20%. We do not take MVGRL (Hassani & Ahmadi, 2020) and SimGRACE (Xia et al., 2022) into comparison because they are two-stage while our comparative methods are one-stage. The settings of our KDGCN-$c$ are consistent with what Appendix A has illustrated while the settings of other methods are their default settings. The results are shown in Table 13, where the least average inference time and memory usage will be highlighted in red.

We can infer from Table 13 that DGCNN consumes the least memory. However, it reaches the highest average inference time. At the same time, GLA has the least average inference time as well as a fairly low memory usage. As for our KDGCN-$c$, it surpasses other methods in mean accuracy by a remarkable margin, while it is not time-consuming. Thus, our transductive KDGCN-$c$ is applicable

and effective on relatively large-scale datasets. We can further reduce our memory usage by utilizing memory collection as well as reducing model parameters (e.g. reducing $d$ and $d'$ in Section A).

Table 12: Time cost of different kernels on PROTEINS, IMDB-M and COLLAB, respectively.

| Kernels | PROTEINS | IMDB-M | COLLAB |
|---|---|---|---|
| shortest path kernel | 0.50 | 0.05 | 7.42 |
| graphlet sampling kernel | 9.82 | 383.42 | - |
| propagation kernel | 1.35 | 0.52 | 41.23 |
| pyramid match kernel | 2.35 | 3.21 | 79.53 |
| neighborhood hash kernel | 1.63 | 0.90 | 58.90 |
| WWL kernel | 6.18 | 6.06 | 324.84 |
| WL subtree kernel | 0.24 | 0.28 | 4.60 |

Table 13: The average inference time and the memory usage for DGCNN, GIN, DropGNN, GLA and our KDGCN-$c$ on PROTEINS, IMDB-M and COLLAB.

| Metrics | Methods | PROTEINS | IMDB-M | COLLAB |
|---|---|---|---|---|
| | DGCNN | 4.08 | 5.86 | 18.59 |
| | GIN | 0.18 | 0.20 | 4.41 |
| average inference time (second) | DropGNN | 0.24 | 0.29 | 7.05 |
| | GLA | 0.13 | 0.14 | 0.64 |
| | KDGCN-$c$(Ours) | 1.39 | 2.14 | 6.98 |
| | DGCNN | 3544.63 | 3551.18 | 3693.49 |
| | GIN | 3636.06 | 3626.18 | 11756.75 |
| memory usage (MB) | DropGNN | 3851.81 | 3847.08 | 4224.29 |
| | GLA | 3694.38 | 3625.78 | 12170.37 |
| | KDGCN-$c$(Ours) | 4277.02 | 4831.28 | 25447.30 |

## F  KDGCN FOR INDUCTIVE LEARNING

Although transductive learning yields satisfactory results in our KDGCN, inductive learning can generalize to unseen nodes and graphs, which broadens the application of our method. Referring to GraphSAGE (Hamilton et al., 2017), we provide an inductive version of our KDGCN, as shown in Algorithm 1, where $t$ is the number of input graphs, $n_j$ is the number of nodes for graph $G_j$, $\mathcal{V}_{n_j}$ is the vertex set for graph $G_j$, $\mathcal{V}_t$ is the vertex set for supergraph $\mathbf{S}$ which consists of $t$ meta-nodes, $\mathcal{N}$ is the neighborhood function related to the adjacency matrix for each graph and $\mathcal{S}$ is the neighborhood function related to the supergraph $\mathbf{S}$. Here, the multilayer perception $g_\varphi$ and the readout function are the same as described in Section 3.2 while the WL subtree kernel and the thresholding operation are the same as described in Section 3.3. For convenience, the dimensions of weight matrices $\mathbf{M}^k$ and $\mathbf{W}^k$ are set to $d \times d$ and the dimensions of the initial node feature vectors are set to $d \times 1$, where $d = 64$.

When training our KDGCN, graphs that are with or without labels are fed into Algorithm 1. Meanwhile, the weight matrices are learnable. When inferring our KDGCN, graphs that are unseen and without labels are fed into Algorithm 1 and matrices that yield the highest validation accuracy are applied. Note that the length of initial node feature vectors are the same for training and inference stage. One can let the initial node feature vector represent the node degree or node label for each node. Due to the time and space limitations, we haven't conducted the corresponding experiments.

---

**Algorithm 1** Forward propagation of inductive KDGCN

---

**Input:** Graph list $\mathcal{G}_{in} = \{G_1, G_2, \ldots, G_t\}$; node feature vectors $\mathbf{x}_j^v, \forall j \in \mathcal{V}_t, \forall v \in \mathcal{V}_{n_j}$; neighborhood depth $K$; weight matrices for neighborhood aggregation $\mathbf{M}^k$ and $\mathbf{W}^k, \forall k \in \{1, 2, \ldots, K\}$; non-linearity $\sigma$; mean aggregation function MEAN; readout function READOUT; thresholding operation THR; WL subtree kernel WL; softmax function SOFTMAX; multilayer perception $g_\varphi$.

**Output:** Label prediction $\hat{\mathbf{Y}}$ of graph list $\mathcal{G}_{in}$.

1: $\mathbf{S} \leftarrow \text{THR}(\text{WL}(\mathcal{G}_{in}))$
2: **for** $j \in \mathcal{V}_t$ **do**
3:      $\mathbf{h}_v^0 \leftarrow \mathbf{x}_j^v, \forall v \in \mathcal{V}_{n_j}$
4:      **for** $k = 1, 2, \ldots, K$ **do**
5:          **for** $v \in \mathcal{V}_{n_j}$ **do**
6:              $\mathbf{h}_v^k \leftarrow \sigma(\mathbf{M}^k \cdot \text{MEAN}(\{\mathbf{h}_v^{k-1}\}) \cup \{\mathbf{h}_w^{k-1}, \forall w \in \mathcal{N}(v))\})$
7:          **end for**
8:          $\mathbf{h}_v^k \leftarrow \mathbf{h}_v^k / \|\mathbf{h}_v^k\|_2, \forall v \in \mathcal{V}_{n_j}$
9:      **end for**
10:      $\mathbf{H}_j \leftarrow [(\mathbf{h}_1^K)^\top, (\mathbf{h}_2^K)^\top, \ldots, (\mathbf{h}_{n_j}^K)^\top]$
11:      $(\bar{\mathbf{h}}_j^0)^\top \leftarrow \text{READOUT}(\mathbf{H}_j)$
12: **end for**
13: **for** $k = 1, 2, \ldots, K$ **do**
14:      **for** $j \in \mathcal{V}_t$ **do**
15:          $\bar{\mathbf{h}}_j^k \leftarrow \sigma(\mathbf{W}^k \cdot \text{MEAN}(\{\bar{\mathbf{h}}_j^{k-1}\}) \cup \{\bar{\mathbf{h}}_a^{k-1}, \forall a \in \mathcal{S}(j))\})$
16:      **end for**
17:      $\bar{\mathbf{h}}_j^k \leftarrow \bar{\mathbf{h}}_j^k / \|\bar{\mathbf{h}}_j^k\|_2, \forall j \in \mathcal{V}_t$
18: **end for**
19: $\hat{\mathbf{Y}} \leftarrow \text{SOFTMAX}(g_\varphi([(\bar{\mathbf{h}}_1^K)^\top, (\bar{\mathbf{h}}_2^K)^\top, \ldots, (\bar{\mathbf{h}}_t^K)^\top]))$

---

