# OpenReview forum: "KDGCN: A Kernel-based Double-level Graph Convolution Network for Semi-supervised Graph Classification with Scarce Labels"
_ICLR.cc/2024/Conference — Submitted to ICLR 2024_

### Official Review · Reviewer_jG7C · 2023-10-25

**Soundness:** 1 poor
**Presentation:** 2 fair
**Contribution:** 1 poor
**Rating:** 3
**Confidence:** 5

**Summary:**

This paper proposes a novel semi-supervised graph classification method that combines GCN modules with graph kernels, resulting in a model with fewer hyperparameters. Experiments on seven benchmark datasets demonstrate its effectiveness compared to various baselines, including supervised GCNs and graph contrastive learning."

**Strengths:**

- Graph classification is a very fundamental problem for graph-related problems, and exploring semi-supervised graph classification is a very interesting topic.
- The paper is well-organized and easy to be understood.

**Weaknesses:**

- The introduction of the graph kernel concept in semi-supervised graph classification methods is not a novel idea, and it has been mentioned in many previous studies [1-3]. However, the authors have not referred to it or provided a detailed comparison, and I strongly recommend that they compare and discuss their work in relation to these existing studies.
- It seems that the graph kernel in the paper is not learnable, which results in the quality of the supergraph construction being entirely dependent on the learned node representations and the chosen threshold. Turning the graph kernel into a learnable component could be a better approach.
- The model is evaluated only on small datasets and doesn't know the scalability on large-scale datasets.
- This task also has several highly relevant works, which the authors have not mentioned or compared to in their paper. To ensure the novelty of their method and the superiority of its results, it is advisable for the authors to provide supplementary comparisons and engage in a detailed discussion. [4-6].

[1] KGNN: Harnessing Kernel-based Networks for Semi-supervised Graph Classification. WSDM 2022

[2] TGNN: A Joint Semi-supervised Framework for Graph-level Classification. IJCAI 2022

[3] GHNN: Graph Harmonic Neural Networks for Semi-supervised Graph-level Classification. Neural Networks 2022

[4] DualGraph: Improving Semi-supervised Graph Classification via Dual Contrastive Learning. ICDE 2022

[5] Active and Semi-supervised Graph Neural Networks for Graph Classification. TBD 2022

[6] Focus on Informative Graphs! Semi-Supervised Active Learning for Graph-Level Classification. 2023

**Questions:**

The novelty of the paper and the absence of important baselines are the two most critical factors affecting the quality of the article. I recommend that the authors make significant revisions.

---

> ### Author Response · Authors · 2023-11-21
> **Authors' rebuttal**
>
> **Response to Weakness 1:**
> Thanks for the comments and references. We were not aware of these related papers when we were preparing our manuscript.
> We are willing to compare these methods in our paper but their code is not publicly available and due to the limited time we are unable to write the code according to the algorithms. We therefore only cited and discussed these important baselines in Section 2 in our updated paper. [1-3] are in Section 2.3, shortly after where we have discussed DropGNN; [4] is in Section 2.4, shortly after where we have discussed GLA; [5-6] are in Section 2.1.
>
> **Response to Weakness 2:**
> Thanks for the suggestion. Currently, the kernels used in our methods are not learnable but as you see, our methods have already outperformed the baselines significantly. The results demonstrated the effectiveness of existing graph kernels such as WL subtree kernel. Indeed, it is possible that learnable kernels can provide better classification performance. We will study this problem in future work and we think the references you provided can give some insights into this problem.
>
> **Response to Weakness 3:**
> In fact, we choose datasets that are commonly used in this field to perform comparison. Our method works effectively on large-scale datasets such as COLLAB, which contains 5000 samples whose number of nodes as well as edges are high. We have also tested REDDIT-B and REDDIT-MULTI5K but owing to the limited time, we have not performed the comparative experiments on these two datasets. Moreover, we have applied several techniques to reduce the complexity and time cost: we have used parallelization when computing the kernel matrix; we have applied sparse matrix multiplication and storage in our GCN modules. Furthermore, we have updated an algorithm which adapts our method to inductive learning, as shown by Algorithm 1 in Appendix F. With the help of inductive learning, our method can better work on large-scale datasets since indutive learning does not require the whole dataset in training and inference.
>
> **Response to Weakness 4:**
> Please refer to our response to Weakness 1.
>
> **Response to Question 1:**
> We'd like to state our novelty in this paper: we design a two-level GCN where the first-level GCN obtains the graph representation $\bar{\mathbf{h}}_i$ and forms the feature vector matrix $\bar{\mathbf{H}}$; the second-level GCN works on the supergraph generated by WL subtree kernel and the feature vector matrix generated by the first-level GCN to obtain the final classification results. Our method is concise in form and fulfills the ability of GCN and WL subtree kernel, achieving satisfactory results on different kinds of datasets. We have updated the classification results regarding $\tau$ and $c$ in Table 2, 3 and 4, respectively. Here, a part of the Table 2 is shown, where we omit the standard deviation as well as the highlight of the second or the third highest accuracy for simplicity:
> \begin{matrix}
> \hline
> \text{method/10\\%}&\text{MUTAG}&\text{PTC-MR}&\text{PROTEINS}&\text{DD}&\text{COLLAB}&\text{IMDB-M}&\text{IMDB-B}\\\\ \hline
> \text{SimGRACE}&83.20&58.32&72.61&74.19&73.87&46.12&68.78\\\\
> \text{GLA}&84.31&56.00&75.04&\textcolor{red}{77.55}&74.84&48.15&68.82\\\\
> \text{KDGCN}-\tau&87.29&61.29&73.85&76.42&74.61&46.27&68.31\\\\
> \text{KDGCN}-c&\textcolor{red}{89.53}&\textcolor{red}{63.16}&\textcolor{red}{76.03}&77.21&\textcolor{red}{88.88}&\textcolor{red}{58.56}&\textcolor{red}{86.22}\\\\\hline
> \text{method/20\\%}&\text{MUTAG}&\text{PTC-MR}&\text{PROTEINS}&\text{DD}&\text{COLLAB}&\text{IMDB-M}&\text{IMDB-B}\\\\
> \hline
> \text{SimGRACE}&86.27&60.58&74.56&76.18&77.57&48.72&71.85\\\\
> \text{GLA}&85.11&58.43&75.34&\textcolor{red}{77.80}&77.84&49.08&72.88\\\\
> \text{KDGCN}-\tau&87.64&62.06&75.18&76.59&77.61&49.60&73.88\\\\
> \text{KDGCN}-c&\textcolor{red}{91.62}&\textcolor{red}{63.81}&\textcolor{red}{76.93}&77.27&\textcolor{red}{90.63}&\textcolor{red}{65.18}&\textcolor{red}{91.70}\\\\
> \hline
> \end{matrix}
> Also, we report the average result across the seven datasets in the following table. It is clear that we have made significant improvements, especially in our KDGCN-c.
> \begin{matrix}
> \hline
> \text{method}&1\\%&2\\%&10\\%&20\\%\\\\ \hline
> \text{WL+SVM}&56.57&59.92&64.80&68.35\\\\
> \text{WL+LP}&54.56&55.87&57.87&58.63\\\\
> \text{DGCNN}&60.52&62.26&65.68&68.62\\\\
> \text{GIN}&59.66&61.39&66.95&69.64\\\\
> \text{DropGNN (Papp et al. NeurIPS 2021)}&57.65&61.49&67.58&69.82\\\\
> \text{SimGRACE (Xia et al. WWW 2022)}&59.30&63.35&68.16&70.82\\\\
> \text{GLA (Yue et al. NeurIPS 2022)}&\textcolor{blue}{61.35}&\textcolor{blue}{63.74}&\textcolor{blue}{69.24}&\textcolor{blue}{70.93}\\\\  \hline
> \text{KDGCN}-\tau&\textcolor{blue}{65.77}&\textcolor{blue}{67.00}&\textcolor{blue}{69.72}&\textcolor{blue}{71.79}\\\\
> \text{KDGCN}-c&\textcolor{red}{70.94}&\textcolor{red}{72.73}&\textcolor{red}{77.08}&\textcolor{red}{79.59}\\\\\hline
> \end{matrix}
>
> We appreciate your comments and are looking forward to your further feedback.

---

> > ### Comment · Reviewer_jG7C · 2023-11-22
> > **Response to authors' rebuttal**
> >
> > Thank you very much for the author's patient and detailed response, but the reply did not alleviate my concerns.
> > - The author mentioned that they were not aware of these works from over a year ago when preparing the manuscript. In my opinion, thorough research should be conducted when delving into a topic. The author compared the latest baseline with methods on graph-level representation learning, rather than the more relevant baselines of semi-supervised graph classification. This is inappropriate. In subsequent improved versions, the author should comprehensively compare these more relevant methods to demonstrate the effectiveness of the proposed approach.
> > - In methods [1,2,3,4], the author consistently introduces the concept of graph kernels for semi-supervised graph classification. Therefore, the incorporation of graph kernel ideas in this paper is not novel. Moreover, in methods [2,3], the introduction of learnable kernels has been demonstrated to enhance the performance of semi-supervised graph classification. Simultaneously, the author's use of graph kernels to construct the 'graph of graphs' concept is quite common, as detailed in reference [5,6].
> > - The author mentions that the graph classification datasets used in the manuscript are common, and I agree with this statement. However, it is even more important to explore the scalability on large-scale datasets, as demonstrated in method [2] using the OGB dataset.
> >
> >  In summary, the innovation in this manuscript falls short of the standards set by ICLR and I will maintain my score unchanged. I hope the author can improve the paper's innovation to better advance the development of semi-supervised graph classification.
> >
> > [1] KGNN: Harnessing Kernel-based Networks for Semi-supervised Graph Classification. WSDM 2022
> >
> > [2] TGNN: A Joint Semi-supervised Framework for Graph-level Classification. IJCAI 2022
> >
> > [3] GHNN: Graph Harmonic Neural Networks for Semi-supervised Graph-level Classification. Neural Networks 2022
> >
> > [4] Focus on Informative Graphs! Semi-Supervised Active Learning for Graph-Level Classification. 2023
> >
> > [5] Imbalanced Graph Classification via Graph-of-Graph Neural Networks. CIKM 2022
> >
> > [6] Few-shot Molecular Property Prediction via Hierarchically Structured Learning on Relation Graphs. Neural Networks 2023

---

> ### Author Response · Authors · 2023-11-22
> **Authors' comments**
>
> **We appreciate your feedback. But we respectfully disagree with you on the three points. And we would like to discuss with you more.**
>
> 1. We are willing to compare the methods of the papers [1-4] you mentioned. But the codes of the four papers are all not publicly available. We have sent emails to the authors of papers [1-4] multiple times (Nov. 11 and Nov. 14) to request the code a few days ago but got no response. We have compared with SimGrace and GLA (published in 2022) which are strong baselines of semi-supervised graph classification. Our method outperformed them significantly.
>
> 2. In methods [1,2,3,4], the authors have introduced the idea of using graph kernels for semi-supervised learning. But this is not a reason that we can never use graph kernels in semi-supervised learning. If yes, three of the four papers [1,2,3,4] should not be published. Simultaneously, if using graph kernels to construct the 'graph of graphs' is quite common, one of the two papers [5,6] should not be published.  Similarly, "using graph kernels to solve semi-supervised graph classification (e.g. label propagation Xiaojin Zhu 2005) is a common idea" means graph kernels shouldn't be used again in semi-supervised graph classification?
>
> More importantly, our method is a double-level GCN model, which is different from the ideas in [1-6]. We found that in [1-4], the improvement of the methods compared to GraphCL and JOAO are limited. In contrast, SimGrace and GLA, both compared in our paper, outperformed GraphCL and JOAO significantly and our method outperformed SimGrace and GLA significantly. This means: **Our methods>SimGrace and GLA>GraphCL and JOAO, methods in [1-4] >= GraphCL and JOAO.** So it can be deduced that the improvement of our method is larger than the improvement of [1-4].
>
> 3. Paper [2] used OGB, which is indeed a large dataset. The largest dataset we used in our paper is the COLLAB dataset, which has 5000 graphs. It should be pointed out that in [1][3][4], the largest dataset is also COLLAB. So why should our paper be rejected?
>
> We'd like to summarize our work using the following words:
> * We proposed a double-level model for semi-supervised graph classification. The model is novel and hasn't been proposed in the literature.
> * Our method is very effective and outperformed strong semi-supervised baselines SimGrace and GLA significantly in almost all cases.
> * We have used a quite large dataset COLLAB in the experiment. We added and discussed the papers mentioned by the reviewer in the revised paper (Section 2.3).

---

> > ### Comment · Reviewer_jG7C · 2023-11-22
> > **Response to authors' rebuttal**
> >
> > The author repeatedly emphasizes the innovation of the paper as the double-level GCN, but this concept has been employed in papers from a long time ago [1] (possibly in the earliest paper on semi-supervised graph classification, although the author does not seem to mention it). The only difference is that the author constructs a 'graph of graphs' using graph kernels. Therefore, the claimed innovation by the author appears to be somewhat insufficient, and the claim by the author that it was not mentioned in the previous literature is not valid.
> >
> > [1] Semi-Supervised Graph Classification: A Hierarchical Graph Perspective. WWW 2019
> >
> > I consistently believe that integrating classic ideas into highly advanced technical concepts is a fantastic approach, but the key lies in how to introduce them in a better and more innovative way. As we all know, ICLR is one of the top conferences in the field of artificial intelligence and machine learning, encouraging authors to develop more novel ideas. The concept of using a double-layer GCN for semi-supervised graph classification has been used by other authors quite early on. If the only modification here is changing the graph construction method, I believe it is a trivial combination and its level of innovation may not meet the standards expected at ICLR.

---

> > > ### Author Response · Authors · 2023-11-23
> > > **Authors' comments**
> > >
> > > Thanks for pointing out the work [1] and discussing with us. We have the following differences:
> > > 1. Different motivations.  Paper [1] considered the case there exists a natural supergraph, while we do not assume it. Our supergraph is based on graph kernels with two effective post-processing techniques.
> > > 2. Different objective functions. Paper [1] learns two classifiers, while we only have one classifier.
> > > 3. Different optimizations. Paper [1] involves mutual information maximization (Jensen–Shannon Divergence) and pseudo-labeling, while our method does not involve these components and hence is very simple.
> > > 4. Different experiments and baselines. Note that our methods have state-of-the-art performance.

---

### Official Review · Reviewer_2nwv · 2023-10-29

**Soundness:** 4 excellent
**Presentation:** 3 good
**Contribution:** 3 good
**Rating:** 8
**Confidence:** 4

**Summary:**

The paper presented a semi-supervised method for graph classification. The proposed model is composed of two GCNs, one is for individual graphs and the other is for a super graph of all graphs, where the super graph is constructed by a graph kernel. The proposed method is compared with its competitors such as graph contrastive learning on benchmark datasets, where different labeling rates have been considered.

**Strengths:**

1. The problem studied in the paper, namely graph-level semi-supervised learning with scarce labels, is an important and challenging problem.
2. The proposed method is based on a double-level GCN model, which has two GCNs. The first one performs graph convolution for each graph and the second one performs graph convolution for a global graph defined (by graph kernel) over all the graphs. This idea is very novel and appealing.
3. The proposed method is compared with state-of-the-art methods such as SimGRACE and GLA as well as classical methods such as GCN and WL kernel. It has competitive performance.
4. The proposed method is simple and easy to implement.

**Weaknesses:**

1. The authors claimed that their method has fewer hyperparameters but they did not provide specific comparison with other methods such as GLA in terms of the number of hyperparameters.
2. The similarity graph among graphs is constructed by a graph kernel such as WL-subtree kernel and there are two different post-processing method for $\mathcal{K}$. it is not clear which one is better and which one was used in the experiments.
3. The writing can be further improved.

**Questions:**

1. At the beginning of Section 3.1, $\mathbf{S}$ is a binary matrix. However, in Section 3.3, the kernel matrix given by a graph kernel may not be binary or sparse. Do the sparsification and binarization have a significant impact on the performance of the proposed method?
2. In Section 4.2, the authors set $d=d’=64$. Is this the best setting? How do $d$ and $d’$ as well as $d’’$ influence the classification accuracy?
3. What are the numbers of layers in the two GNNs in the experiments? Does the depth matter?
4. In Figure 2, the two post-processing methods for the global kernel matrix are compared. It seems that the one related to $c$ is better than the one related to $\tau$. I wonder if the authors reported the results of the method related to $c$ in Tables 2, 3, and
5. It is not clear why the authors did not include the results of larger labeling rates such as 30% or 50%.
6. Are their any time cost comparison?
7. In Table 4, it seems that the performance of graphlet sampling kernel is always the worst. I suggest the authors discuss the difference between graphlet sampling kernel and other kernels.
8. It is necessary to compare the number of hyperperameters of the proposed method with those of the baselines. In the proposed method, one has to determine $c$ or $\tau$, which affect the classification performance.

---

> ### Author Response · Authors · 2023-11-21
> **Authors' rebuttal**
>
> **Response to Weakness 1:**
> In fact, merely tuning $\tau$ or $c$ in our method can ensure high classification accuracy. Other hyperparameters including learning rate and the number of hidden units ($d, d'$ and $d''$) are chosen empirically (Many papers choose 16 or 64 for the number of hidden units). As for the graph contrastive learning methods such as GLA (Yue et al., 2022), different kinds of data augmentation techniques are needed. In GLA, there are at least two hyperparameters regarding which data augmentation technique to choose and two hyperparameters regarding the augmentation ratio. Moreover, the $\eta$ in GLA, which determines the permutation in augmentation, is also an important hyperparameter to tune. Furthermore, these graph contrastive learning methods often yield complex model architecture, which needs various kinds of layers of different depths. In contrast, our method is concise in form and does not need to tune the hyperparameters regarding the number of layers. Additionally, the learning rate and the epoch are the same for all datasets in our method, while some methods need specific settings for different datasets.
>
> As for other important baselines, DGCNN (Zhang et al., 2018) needs to tune the pooling rate and it sets different learning rates as well as epochs for different datasets; GIN (Xu et al., 2019) needs to tune the number of layers as well as the number of MLP layers; MVGRL  (Hassani & Ahmadi, 2020) needs to tune the number of layers as well as the early stopping steps; SimGRACE (Xia et al., 2022) needs to tune the number of layers as well as the perturbation $\eta$.
>
> **Response to Weakness 2:**
> Thanks for pointing it out. In Table 2 and 3, we have updated the results regarding $\tau$ and $c$, respectively. Results show that the thresholding operation regarding $c$ is better than that regarding $\tau$ in most cases. We have also analyzed this in Section 4.3 and Appendix D.
>
> **Response to Question 1:**
> Yes, the binarization is significant since the binarized kernel matrix is consistent with the adjacency matrix of a single graph in form. Then, the binarized kernel matrix can serve as the “adjacency matrix” among graphs (i.e. meta-nodes). Also, since WL subtree kernel may be tolerant to two graphs that are actually not structurally isomorphic, a binarization can ensure the effectiveness of the kernel. We have tested that our method may not converge when we do not perform binarization.
>
> **Response to Question 2:**
> This is an empirical setting and it yields fine results. $d, d'$ and $d''$ that are too high or too low may lead to sub-optimal results. Detailed comparison of different $d, d'$ and $d''$ is demonstrated in Table 6 and 7 in Appendix B.
>
> **Response to Question 3:**
> The number of layers is 2, which is consistent with the original GCN settings (Kipf & Welling, 2017). It has been proven by GCN that if the number of layers is too high, the performance of GCN module will decline.
>
> **Response to Question 4:**
> In Tables 2, 3 and 4, we have updated the results regarding $\tau$ and $c$, respectively. A part of the updated Table 2 is shown below, where we omit the standard deviation and the highlight for simplicity:
> \begin{matrix}
> \hline
> \text{method/10\\%}&\text{MUTAG}&\text{PTC-MR}&\text{PROTEINS}&\text{DD}&\text{COLLAB}&\text{IMDB-M}&\text{IMDB-B}\\\\ \hline
> \text{SimGRACE}&83.20&58.32&72.61&74.19&73.87&46.12&68.78\\\\
> \text{GLA}&84.31&56.00&75.04&77.55&74.84&48.15&68.82\\\\
> \text{KDGCN}-\tau&87.29&61.29&73.85&76.42&74.61&46.27&68.31\\\\
> \text{KDGCN}-c&89.53&63.16&76.03&77.21&88.88&58.56&86.22\\\\\hline
> \text{method/20\\%}&\text{MUTAG}&\text{PTC-MR}&\text{PROTEINS}&\text{DD}&\text{COLLAB}&\text{IMDB-M}&\text{IMDB-B}\\\\
> \hline
> \text{SimGRACE}&86.27&60.58&74.56&76.18&77.57&48.72&71.85\\\\
> \text{GLA}&85.11&58.43&75.34&77.80&77.84&49.08&72.88\\\\
> \text{KDGCN}-\tau&87.64&62.06&75.18&76.59&77.61&49.60&73.88\\\\
> \text{KDGCN}-c&91.62&63.81&76.93&77.27&90.63&65.18&91.70\\\\
> \hline
> \end{matrix}
>
> **Response to Question 5:**
> We think that a low label rate can better testify the ability of our method to extract structural information for semi-supervised classification. Also, we have explored some papers that apply low label rates, including SimGRACE (Xia et al., 2022).
>
> **Response to Question 6:**
> In Appendix E, we have compared the time cost of different kernels in Table 12 as well as different graph classification methods in Table 13. Results show that WL subtree kernel is efficient and our method is not time-consuming.
>
> **Response to Question 7:**
> In our updated Table 4, graphlet sampling kernel with thresholding regarding $c$ performs the best on PROTEINS and IMDB-M. However, it is very time-consuming since it has to generate lots of graphlets of different sizes. We have discussed why we choose WL kernel in Section 4.5.
>
> **Response to Question 8:**
> Please refer to our response to Weakness 1.
>
> We appreciate your comments and are looking forward to your further feedback.

---

### Official Review · Reviewer_Qkvr · 2023-10-31

**Soundness:** 3 good
**Presentation:** 2 fair
**Contribution:** 2 fair
**Rating:** 5
**Confidence:** 3

**Summary:**

- The paper studies the problem of graph classification with scarce labels. The authors propose a semi-supervised graph classification method called KDGCN, which consists of two GCN modules. The first GCN module obtains feature vectors for each graph through a readout operation. Then, the authors construct a supergraph using graph kernels. The second GCN module employs a semi-supervised approach to learn meta-node representations on the supergraph, capturing sufficient structural information from both labeled and unlabeled graphs. Typically, semi-supervised methods based on graph contrastive learning result in complex models and intricate hyperparameter-tuning. However, the method proposed by the authors has fewer hyperparameters and is easy to implement.

**Strengths:**

- The paper is overall easy to understand.
- The idea of constructing a supergraph is novel and interesting.
- When graph labels are extremely scarce, the proposed method has shown some improvements on certain datasets.

**Weaknesses:**

- The section about supergraph construction mentions using a predefined similarity threshold (τ) to determine the existence of edges, but it does not explain how to select this threshold.
- While the experiments demonstrate that the WL subtree kernel performs well in certain cases, should the paper provide a more detailed comparison and analysis to explain why this kernel was chosen over other possible kernels?

**Questions:**

- Can more information be provided to explain the structure and properties of the supergraph and how it impacts the method's performance?
- I am concerned about the limitations of the proposed method and its potential application scenarios. Additionally, is the complexity of the proposed method scalable on large datasets?

---

> ### Author Response · Authors · 2023-11-21
> **Authors' rebuttal**
>
> **Response to Weakness 1:**
> Thanks for pointing it out. This threshold is selected via grid-searching. Intuitively, if $\tau$ is too large or $c$ is too small in the thresholding operation in Section 3.3, the supergraph will be too sparse. However, if $\tau$ is too small or $c$ is too large, the supergraph will be a very dense graph and is not discriminative, which leads to low classification accuracy. Thus, we can perform the grid-search within a moderate range to find the threshold. The discussion is added to Section 3.3. Furthermore, we have updated the classification results of our method using different thresholding operations in Table 2 and 3. A part of the Table 2 is shown below, where we omit the standard deviation and the highlight of the second or the third highest accuracy for simplicity:
> \begin{matrix}
> \hline
> \text{method/10\\%}&\text{MUTAG}&\text{PTC-MR}&\text{PROTEINS}&\text{DD}&\text{COLLAB}&\text{IMDB-M}&\text{IMDB-B}\\\\ \hline
> \text{GIN}&82.35&56.52&71.94&70.14&73.04&47.50&67.16\\\\
> \text{DropGNN}&82.00&57.61&72.77&75.19&69.71&46.36&69.43\\\\
> \text{SimGRACE}&83.20&58.32&72.61&74.19&73.87&46.12&68.78\\\\
> \text{GLA}&84.31&56.00&75.04&\textcolor{red}{77.55}&74.84&48.15&68.82\\\\
> \text{KDGCN}-\tau&87.29&61.29&73.85&76.42&74.61&46.27&68.31\\\\
> \text{KDGCN}-c&\textcolor{red}{89.53}&\textcolor{red}{63.16}&\textcolor{red}{76.03}&77.21&\textcolor{red}{88.88}&\textcolor{red}{58.56}&\textcolor{red}{86.22}\\\\\hline
> \text{method/20\\%}&\text{MUTAG}&\text{PTC-MR}&\text{PROTEINS}&\text{DD}&\text{COLLAB}&\text{IMDB-M}&\text{IMDB-B}\\\\
> \hline
> \text{GIN}&85.36&60.90&74.07&70.71&75.54&48.65&72.22\\\\
> \text{DropGNN}&82.71&61.63&74.14&75.79&71.66&50.05&72.79\\\\
> \text{SimGRACE}&86.27&60.58&74.56&76.18&77.57&48.72&71.85\\\\
> \text{GLA}&85.11&58.43&75.34&\textcolor{red}{77.80}&77.84&49.08&72.88\\\\
> \text{KDGCN}-\tau&87.64&62.06&75.18&76.59&77.61&49.60&73.88\\\\
> \text{KDGCN}-c&\textcolor{red}{91.62}&\textcolor{red}{63.81}&\textcolor{red}{76.93}&77.27&\textcolor{red}{90.63}&\textcolor{red}{65.18}&\textcolor{red}{91.70}\\\\
> \hline
> \end{matrix}
>
> **Response to Weakness 2:**
> In Section 4.5, we have updated a discussion on why we choose WL subtree kernel over other kernels. First, the time complexity of WL subtree kernel is linear with the number of edges of the graph and hence is more efficient than many other kernels. Second, the WL subtree kernel often yields better and stabler classification results than other kernels.
>
> **Response to Question 1:**
> Firstly, we added a study in Appendix D on the structural properties of our supergraphs constructed via $\tau$ and $c$, respectively. These properties include average node degree, number of connected components, average clustering coefficient, average degree centrality, average closeness centrality and the top smallest eigenvalues of the Laplacian matrices of our supergraphs. On PROTEINS, we have investigated how the properties change with $\tau$ and $c$, respectively. With the rise of $\tau$, or with the fall of $c$, the average node degree and the average closeness centrality decline, indicating that the supergraph becomes sparser. Referring to Figure 3(a) and Figure 3(b) in Appendix B, a sparse supergraph is sub-optimal to our method. Besides, we explore the mean accuracy results on different datasets when $\tau$ or $c$ changes, as shown in Figure 2, 3 and 4 in Appendix B.
>
> We further analyze that supergraphs constructed via $c$ enjoy fair connectivity as well as certain centrality, which ensure the message passing as well as emphasize important meta-nodes. However, supergraphs constrcuted via $\tau$ are sparse, centralized and disconnected, resulting in less effective message passing. Thus, we have proven that the thresholding operation regarding $c$ is better than that regarding $\tau$, especially on social networks datasets.
>
> **Response to Question 2:**
> Firstly, our kernel matrix (i.e. supergraph) is pre-computed and will not result in additional complexity during training and inference stages. The complexity when obtaining the supergraph can be further reduced via parallelization. Moreover, we apply GCN modules in our method, which do not contain abundant parameters. Furthermore, we apply sparse matrix techniques on storage and computation in our method, lowering the complexity as well as the time cost. A commonly-used GPU platform such as RTX 2080Ti is sufficient for our method on large-scale datasets such as COLLAB.
> It is worth noting that our method can be easily extended to graph attention based models such as GAT [Veličković et al. 2018] or inductive graph learning such as GraphSAGE [Hamilton et al. 2017], which are more effective in handling large datasets. Please take a look at our Algorithm 1 in the appendix. It is the extension to inductive learning.
>
> We appreciate your comments and are looking forward to your further feedback.

---

### Official Review · Reviewer_mRm5 · 2023-11-01

**Soundness:** 3 good
**Presentation:** 2 fair
**Contribution:** 2 fair
**Rating:** 5
**Confidence:** 4

**Summary:**

This paper views graphs as meta-nodes and constructs a super graph, which then enables semi-supervised graph classification learning, akin to semi-supervised node classification learning. Specifically:

1. First, a GNN is used to learn a representation for each graph, serving as the initial node representation of the supergraph,
2. Next, the WL kernel is employed to determine the similarity between graphs, forming the edges of the supergraph,
3. Finally, another GNN is used for semi-supervised learning on the supergraph.

The experiments implied that this method can achieve SOTA or comparable to SOTA results on several datasets.

**Strengths:**

1. Compared to other methods based on contrastive learning, utilizing a supergraph for semi-supervised learning eliminates the need to construct negative samples, simplifying the whole framework.

2. It achieves SOTA results on smaller datasets and comes close to SOTA on medium-sized datasets.

**Weaknesses:**

1. The datasets used for experiments are relatively small, and it seems that the advantages are not as pronounced on larger datasets, necessitating validation on larger datasets.

2. A comparison is needed with the following two papers:

    [1]. **Few-Shot Learning on Graphs via Super-Classes based on Graph Spectral Measures**

    [2]. **PRODIGY: Enabling In-context Learning Over Graphs**

In paper [a], a supergraph is constructed for Few-Shot graph classification, while in paper [b], a supergraph is built for In-context few-shot node and *edge classification*.

**Questions:**

1. This paper mentions that the two GCNs are optimized jointly, implying that during training, all graphs in the dataset must be inputted into the hardware simultaneously. Does this limit the model's ability to be trained on large-scale datasets?

2. If KDGCN only supports the Transductive setting, while the compared methods MVGRL, SimGRACE, and GLA can support the Inductive setting?

3. If it is the Transductive setting, must the entire dataset be inferred together during inference? Please describe the inference budget, including platform, memory usage, and inference time.

4. Is this paper the first to perform semi-supervised graph classification by constructing a supergraph? The core innovative point of the article needs to be re-emphasized.

---

> ### Author Response · Authors · 2023-11-21
> **Authors' rebuttal**
>
> **Response to Weakness 1:**
> We choose datasets that are commonly used in this field to perform comparisons. Meanwhile, COLLAB which contains 5000 samples is quite large-scale, where the average number of nodes as well as edges are also large.
>
> In the revised paper, we supplement the experiments of our method KDGCN with the second post-processing method for graph kernel matrix, named KDGCN-c. As shown by Table 2 and Table 3, KDGCN-c outperformed all baselines in 26 out of 28 cases.
>
> We report the average result across the seven datasets in the following table. We see that the improvement of our methods especially KDGCN-c is remarkable.
>
> \begin{matrix}
> \hline
> \text{method}  &1\\% &2\\% &10\\% &20\\% \\\\ \hline
> \text{WL+SVM} &56.57 &59.92 &64.80 &68.35\\\\
> \text{WL+LP} &54.56 &55.87 &57.87 &58.63\\\\
> \text{DGCNN} &60.52 &62.26 &65.68 &68.62\\\\
> \text{GIN} &59.66 &61.39 &66.95 &69.64\\\\
> \text{DropGNN (Papp et al., NeurIPS 2021) } &57.65 &61.49 &67.58 &69.82\\\\
> \text{SimGRACE (Xia et al., WWW 2022) }&59.30 &63.35 &68.16 &70.82\\\\
> \text{GLA (Yue et al., NeurIPS 2022)}&\textcolor{blue}{61.35} &\textcolor{blue}{63.74} &\textcolor{blue}{69.24} &\textcolor{blue}{70.93}\\\\      \hline
> \text{KDGCN}-\tau &\textcolor{blue}{65.77} &\textcolor{blue}{67.00} &\textcolor{blue}{69.72} &\textcolor{blue}{71.79}\\\\
> \text{KDGCN}-c &\textcolor{red}{70.94} &\textcolor{red}{72.73} &\textcolor{red}{77.08} &\textcolor{red}{79.59}\\\\ \hline
> \end{matrix}
>
> **Response to Weakness 2**:
> Thanks for pointing out the references. We have cited and discussed these papers in the second paragraph of Section 2.2. However, it is hard to apply the methods of the two papers to the semi-supervised graph-level classification tasks. The reasons are as follows:
> 1. The first paper is about few-shot learning, which is quite different from our topic. By the way, the paper was published in 2020 and we have compared three newer methods including DropGNN (Papp et al., 2021), SimGRACE (Xia et al., 2022) and
> GLA (Yue et al., 2022).
> 2. The second paper (PRODIGY) studies the problem of in-context learning and pre-trained model, which is different from the problem studied in our work. The method PRODIGY proposed in the reference mainly aims at node classification or relation prediction, but our method aims at graph-level classification.
>
> **Response to Question 1**:
> This is an insightful question. Yes, all graphs are inputted into the storage but they are in the form of sparse matrix, which does not have high memory cost and also makes the computation of matrix multiplications efficient. For instance, our method can be efficiently applied to the COLLAB dataset that is quite large.
>
> We reported the time cost comparison in Table 13. We can see that our method is more efficient than DGCNN and as fast as DropGNN.
>
> It is worth noting that our method can be easily extended to graph attention based models such as GAT (Veličković et al., 2018) or inductive graph learning such as GraphSAGE (Hamilton et al., 2017), which are more effective in handling large datasets.
>
> **Response to Question 2**:
> Thanks for the insightful question. Currently, because our KDGCN is based on GCN modules, it cannot be directly applied to inductive learning. But as mentioned in the previous question, we can easily adapt KDGCN to inductive learning, by replacing graph convolution with the neighbor aggregation used in GraphSAGE. The algorithm is shown by Algorithm 1 in the appendix.
>
> **Response to Question 3**:
> Yes, but as mentioned before, the computation cost is not very high as the graphs are in the form of sparse matrices.
> Also, a commonly-used platform with GPU RTX 2080Ti is sufficient for our method to operate on. The inference time and the total memory usage are listed in Table 13. The theoretical analysis of time and space complexities are in Section 3.4.
>
> **Response to Question 4**:
> Ju et al. (2022a;b;a), as mentioned by Reviewer jG7C, have raised methods that construct supergraphs. For example, Chauhan et al. (2020) cluster prototype graphs based on Wasserstein spectral distance to form the supergraph; Ju et al. (2022) used graph kernel to assist message passing and build up an auxiliary probability distribution for unlabeled graphs. Differently, our method directly applies WL subtree kernel to generate a similarity supergraph, which is a more direct guide for classification since WL subtree kernel can significantly capture the isomorphism among graphs. Meanwhile, we incorporate WL subtree kernel into GCN modules of different levels. The first-level GCN module is to obtain graph representation $\bar{\mathbf{h}}_i$ and formulate the “feature vector matrix” for the supergraph while the second-level GCN module is to obtain the final classification result. Our methods are very different from the related papers. The double-level GCN is novel.
>
> We appreciate your comments and are looking forward to your further feedback.

---

### Meta-Review · Area_Chair_BHsE · 2023-12-09

**Metareview:**

The paper proposes to use double-level GCN to do graph classification. A WL subtree kernel is used to construct a meta-graph among all graphs in the dataset, where edges are determined by kernel similarity score. The first-level GCN computes graph embedding first for each graph as the node embedding in the super-graph, then the second-level GCN performs semi-supervised node classification on the super-graph, achieving graph classification. The two-level GCNs are optimized jointly. The idea is novel and interesting. However, the main concerns lie in the transductive setting where all graphs in the dataset must be loaded together. This is not feasible for large-scale datasets. The authors also only used TU datasets without using any modern large-scale datasets (such as OGB, zinc, QM9) for evaluation. The feasibility to only small datasets significantly limit its applicability. Furthermore, the two-level GCN framework based on graph kernel seems a heuristic approach. The WL subtree information can be fully preserved in GNN embeddings, so using another WL kernel to compute similarity graph seems to introduce no more information than the first GCN itself. Overall, the paper needs to test the proposed method on larger datasets and provide more theoretical justifications.

**Justification For Why Not Higher Score:**

Only test on small TU datasets.

**Justification For Why Not Lower Score:**

N/A

---

### Decision · Program_Chairs · 2024-01-16

Reject